# In situ formation of ZnO$_x$ species for efficient propane dehydrogenation

Dan Zhao[1,2], Xinxin Tian[2,3], Dmitry E. Doronkin[4], Shanlei Han[1,2], Vita A. Kondratenko[2], Jan-Dierk Grunwaldt[4], Anna Perechodjuk[2], Thanh Huyen Vuong[2], Jabor Rabeah[2], Reinhard Eckelt[2], Uwe Rodemerck[2], David Linke[2], Guiyuan Jiang[1 ✉], Haijun Jiao[2 ✉] & Evgenii V. Kondratenko[2 ✉]

Propane dehydrogenation (PDH) to propene is an important alternative to oil-based cracking processes, to produce this industrially important platform chemical[1,2]. The commercial PDH technologies utilizing Cr-containing (refs. [3,4]) or Pt-containing (refs. [5–8]) catalysts suffer from the toxicity of Cr(vi) compounds or the need to use ecologically harmful chlorine for catalyst regeneration[9]. Here, we introduce a method for preparation of environmentally compatible supported catalysts based on commercial ZnO. This metal oxide and a support (zeolite or common metal oxide) are used as a physical mixture or in the form of two layers with ZnO as the upstream layer. Supported ZnO$_x$ species are in situ formed through a reaction of support OH groups with Zn atoms generated from ZnO upon reductive treatment above 550 °C. Using different complementary characterization methods, we identify the decisive role of defective OH groups for the formation of active ZnO$_x$ species. For benchmarking purposes, the developed ZnO–silicalite-1 and an analogue of commercial K–CrO$_x$/Al$_2$O$_3$ were tested in the same setup under industrially relevant conditions at close propane conversion over about 400 h on propane stream. The developed catalyst reveals about three times higher propene productivity at similar propene selectivity.

Encouraged by the requirements of environmental sustainability, both industry and academia are searching for ecologically friendly PDH catalysts. The most promising materials are based on oxides of V (refs. [10,11]), Ga (ref. [12]), Zr (refs. [13,14]) or Co (refs. [15,16]). Zn-containing catalysts were also tested but show commercially unattractive performance[17–27]. A general shortcoming of practically all supported catalysts developed until now is their economic inefficiency due to the complex preparation methods often requiring expensive chemicals. In addition, the structure of active supported species can undergo irreversible reaction-induced changes causing an unalterable catalyst deactivation.

Against the above background, our intention was to offer a preparation method with a potential of large-scale applications using commercially available ZnO and oxidic supports to yield catalysts with industrially relevant activity, durability and selectivity. In contrast to previous relevant studies[17–27], we simply tested a physical mixture of ZnO (8wt%) and non-acidic SiO$_2$ material (silicalite-1 (S-1_1), silicalite-2 (S-2), MCM-41, SBA-15 or non-structured SiO$_2$) to hinder undesired side reactions. ZnO and the bare supports tested separately are practically inert. The activity of the physical mixtures depends on the kind of support and pre-treatment (Fig. 1a). Propane conversion ($X$(C$_3$H$_8$)) over oxidized ZnO-S-1_1 increases from 6% to 24% within 1.5 h on propane stream, while high conversion of about 30% is achieved directly after treatment in H$_2$/N$_2$ = 1 at 550 °C for 1 h before PDH (Extended Data Fig. 1a). Other either oxidized or reduced mixtures show low activity

(Fig. 1a). Thus, when testing ZnO-S-1_1, catalytically active species must be formed upon interaction between ZnO and S-1_1 under reaction conditions or upon reductive treatment. What are these sites and how are they formed?

To answer these questions, we prepared a series of additional catalysts. ZnO and S-1_1, S-2 or MCM-41 were loaded into a tubular reactor in form of two layers separated by a quartz wool with the upstream layer being ZnO (inset in Fig. 1b). After exposing to H$_2$/Ar = 1 at 550 °C for 1 h and finally cooling down to room temperature in Ar, the downstream support-containing layer was collected and abbreviated as S-1_1(H$_2$), S-2(H$_2$) and MCM-41(H$_2$). Although zinc is present in these materials in similar amounts, S-1_1(H$_2$) (1.2wt% Zn) shows higher propene formation rate ($r$(C$_3$H$_6$)) than MCM-41(H$_2$) (0.98wt% Zn) and S-2(H$_2$) (2.4wt% Zn) (Fig. 1b and Extended Data Fig. 1b). S-1_1(O$_2$) or S-1_1(Ar) prepared in 5vol% O$_2$/Ar or Ar do not contain zinc and are inactive (Fig. 1b).

As zinc is present in S-1_1(H$_2$), S-2(H$_2$) and MCM-41(H$_2$), there must be a gas-phase transport of Zn-containing species to the supports because ZnO and support layers were separated by a quartz wool. To check if Zn$^0$ or ZnO is transported, we heated a physical mixture of Zn$^0$ and S-1_1 in Ar. H$_2$ evolution was observed between 300 and 400 °C (Fig. 1c). Thus, H$_2$ should originate from the oxidation of Zn$^0$ by OH groups to yield supported ZnO$_x$. In situ DRIFTS (diffuse reflectance infrared Fourier transform spectroscopy) measurements prove that OH groups preferentially react with Zn$^0$. When treating a ZnO–S-1_1 mixture in H$_2$/Ar = 1

[1]State Key Laboratory of Heavy Oil Processing, China University of Petroleum, Beijing, P. R. China. [2]Leibniz-Institut für Katalyse e.V., Rostock, Germany. [3]Key Laboratory of Materials for Energy Conversion and Storage of Shanxi Province, Institute of Molecular Science, Shanxi University, Taiyuan, P. R. China. [4]Institute of Catalysis Research and Technology and Institute for Chemical Technology and Polymer Chemistry, Karlsruhe Institute of Technology (KIT), Karlsruhe, Germany. ✉e-mail: jiangy@cup.edu.cn; haijun.jiao@catalysis.de; evgenii.kondratenko@catalysis.de

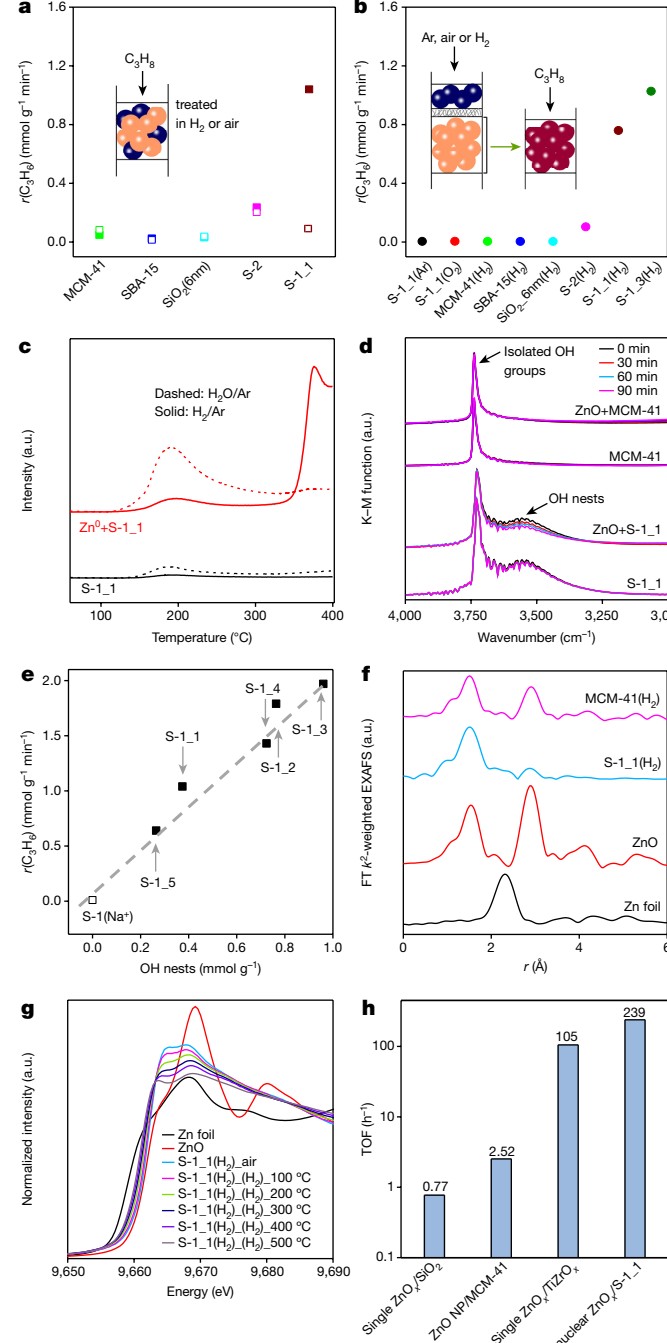

**Fig. 1 | Catalyst activity, formation of ZnO$_x$ species and their structure.**
**a**, $r(C_3H_6)$ at 550 °C over oxidatively (open symbols) or reductively (closed symbols) treated mixtures of ZnO (blue spheres) and SiO$_2$-based support (orange spheres). **b**, $r(C_3H_6)$ over Zn-containing materials (dark-red spheres), prepared as shown in the inset. **c**, H$_2$ and H$_2$O evolution upon temperature-programmed heating of a mixture of Zn$^0$ and S-1_1 in Ar. **d**, In situ DRIFTS spectra of S-1_1, MCM-41, or their mixtures with 10wt% ZnO at 550 °C in H$_2$/Ar = 1. **e**, $r(C_3H_6)$ over reduced ZnO-silicalite-1 at 550 °C versus the concentration of OH nests in differently prepared silicalite-1. **f**, Ex situ FT EXAFS spectra (not corrected for the phase shift) of different materials. **g**, In situ XANES spectra of S-1_1(H$_2$) treated in 5vol% H$_2$ in He at different temperatures. **h**, Turnover frequency (TOF) of propene formation over Zn-based catalysts. The values of single ZnO$_x$/SiO$_2$ and single ZnO$_x$/TiZrO$_x$ are reported in ref. [19] and ref. [24], respectively.

at 550 °C, the intensity of band at 3,540 cm$^{-1}$ characteristic[28] for such defects in S-1_1 decreases with time on H$_2$ stream (Fig. 1d). No such changes were observed upon reduction of the sole supports or a mixture of ZnO and MCM-41 possessing exclusively isolated OH groups.

The importance of OH nests for creation of ZnO$_x$ species is demonstrated further. In addition to S-1_1, S-1 materials (S-1_2, S-1_3, S-1_4 and S-1_5) with different concentration of OH nests were synthesized and characterized (Extended Data Table 1 and Extended Data Fig. 1c–h). When each of them was separately tested as a physical mixture with ZnO, a positive linear correlation between $r(C_3H_6)$ and the number of OH nests on the support was established (Fig. 1e). A mixture of ZnO and Na-containing S-1 with exclusively isolated OH groups shows very low $r(C_3H_6)$ (Extended Data Fig. 1i). OH nests appear after removal of Na$^+$ and the rate increases by about 14 times. ZnO$_x$ species formed in these materials should be of same structure because a similar apparent activation energy of propene formation was determined for ZnO–S-1 mixtures (Extended Data Fig. 1j).

To determine the local structure of ZnO$_x$ species, the catalysts were prepared according to the inset in Fig. 1b and characterized by X-ray absorption spectroscopy. The oxidation state of Zn in S-1_1(H$_2$) is +2 as the absorption edge in the X-ray absorption near-edge structure (XANES) spectrum is close to 9,662 eV characteristic for ZnO. Partially reduced ZnO$_x$ (defect structure with oxygen vacancies) should be present in MCM-41(H$_2$) (Extended Data Fig. 2a, b). As concluded from the Fourier transformed $k^2$-weighted extended X-ray absorption fine structure (EXAFS) spectra of these catalysts (Fig. 1f), the first and the second shell scatterings at 1.5 and 2.9 Å correspond to O and Zn neighbours as in ZnO, respectively. All fitting details are summarized in Extended Data Figs. 2e–j and 3, and Extended Data Table 2. Based on the average coordination number (CN) in Zn–O of 2.5 and in Zn–Zn of 8.4, MCM-41(H$_2$) should contain ZnO$_x$ nanoparticles. The corresponding CN values for S-1_1(H$_2$) are 3 and 1. Thus, S-1_1(H$_2$) should contain binuclear ZnO$_x$ species. Using Si instead of Zn is not able to satisfactorily predict the second shell (Extended Data Fig. 2g–j). We also explored if and how the structure of such species in S-1_1(H$_2$) changes upon catalyst treatment in H$_2$ to mimic its state in PDH. The position of the Zn K-absorption edge shifts to lower energy upon heating from 100 to 500 °C (Fig. 1g). In addition, CN in the first Zn–O shell decreases from 3 to 2. The intensity in the EXAFS spectra at 1.5 Å decreases too (Extended Data Fig. 2c, d). Contrarily, the intensity in the EXAFS spectra at 2.9 Å decreases only slightly upon heating to 200 °C and does not change with a further temperature rise. Thus, no changes in CN of Zn–Zn occur upon catalyst reductive treatment.

The ability of binuclear ZnO$_x$ species to lose lattice oxygen upon reductive treatment was independently proven by O$_2$-titration tests with S-1_1(H$_2$) treated in H$_2$ at 550 °C. O$_2$ was consumed until all vacancies were filled (Extended Data Fig. 4a). These defects were also directly identified by in situ electron paramagnetic resonance spectroscopy as F$^+$ centres at a $g$ factor of 2.005 characteristic for a trapped electron in an oxygen vacancy[29]. Their concentration increases after reducing S-1_1(H$_2$) in H$_2$/Ar = 1 at 550 °C (Extended Data Fig. 4b). No F$^+$ centres could be seen after reductive treatment of bare S-1_1 (Extended Data Fig. 4c).

Based on the above discussion, the active site for PDH over S-1_1(H$_2$) should be lower coordinated binuclear ZnO$_x$ species. It reveals about 300, 2 or 100 times higher turnover frequency of propene formation in comparison with isolated ZnO$_x$ species present in ZnO$_x$/SiO$_2$ and ZnO$_x$/TiZrO$_x$ or ZnO$_x$ nanoparticles in ZnO$_x$/MCM-41 (Fig. 1h). Moreover, the developed catalysts also show higher propene productivity under industrially relevant conditions in comparison with the state-of-the-art Zn-containing catalysts (Extended Data Table 3).

Kinetically relevant step(s) in the PDH reaction were investigated using the temporal analysis of products (TAP) reactor operating with sub-millisecond resolution[30]. C$_3$H$_6$ or C$_3$D$_6$ were detected upon pulsing of C$_3$H$_8$/Ar = 1 or C$_3$D$_8$/Ar = 1 over S-1_1(H$_2$) at 550 °C (Extended Data

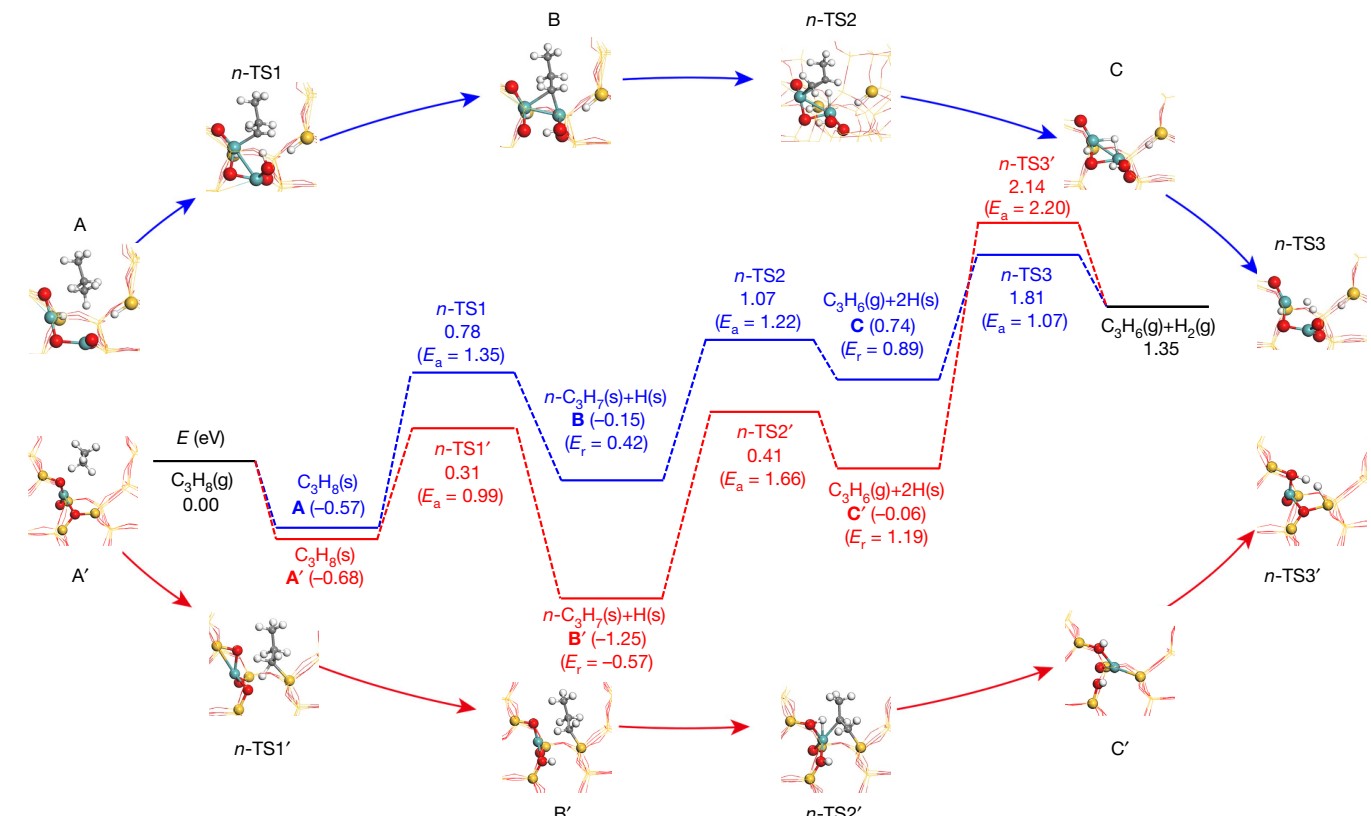

**Fig. 2 | Molecular details of PDH.** The calculated energy profiles along the minimum energy pathways and the optimized intermediates (A–C) and transition states (TS) at the PBE + D3 + ZPE level upon PDH on the reduced binuclear $ZnO_x$ (blue) and single $ZnO_x$ (red) sites on the surface of S-1. Cyan, red, grey, yellow and white spheres stand for Zn, O, C, Si and H atoms, respectively.

Fig. 4d, e). The amount of $C_3D_6$ was significantly lower. Neither $H_2$ nor $D_2$ were detected. The reason for their absence might be the lower sensitivity of mass spectrometric analysis toward these products in comparison with propene. Alternatively, when $H_2$ was formed very slowly, the ratio of signal to noise would be too low for proper detection. To check if $H_2$ formation limits the PDH reaction, H/D exchange tests were carried out at 550 °C using a $D_2$/Ne = 1 mixture. When this mixture was pulsed over S-1_1($H_2$), HD was observed while this product was not formed in tests with bare S-1_1 and MCM-41($H_2$) (Extended Data Fig. 4f–h). Thus, binuclear $ZnO_x$ species in S-1_1($H_2$) must be involved in the exchange reaction. D atoms formed from $D_2$ on these species react with OH groups of S-1_1($H_2$) to yield HD. This process should be fast because the initial conversion of $D_2$ is high (Extended Data Fig. 4i). The conversion decreases with rising number of $D_2$ pulses due to the transformation of OH groups into OD groups. The high activity to HD exchange and the lower amount of propene formed from deuterated propane suggest that the breaking of C−H bonds in propane should be the rate-limiting step in PDH.

Molecular-level details of PDH were derived from DFT calculations. A model for binuclear $ZnO_x$ species was created according to our characterization data mentioned above. Two Si defects were substituted by two Zn atoms in the orthorhombic MFI framework (Extended Data Fig. 5a, b). The positively charged Zn atoms are 2O- and 3O-coordinated. We considered breaking of methylene or methyl C−H bonds in $C_3H_8$ and successive transformations of the formed iso-$C_3H_7$ or n-$C_3H_7$ intermediates to gas-phase $C_3H_6$ and $H_2$ (Extended Data Figs. 5d and 6). The pathway starting with the heterolytic methyl C−H bond cleavage seems to be kinetically relevant. For the most preferred route (Fig. 2), the first C−H bond cleavage occurs on the low-coordinated $Zn^{\delta+}$ with a barrier of 1.35 eV. The formed n-$C_3H_7$ is in a bridged position between two $Zn^{\delta+}$

after substituting one Zn−O bond of the 3O-coordinated $Zn^{\delta+}$ by Zn−Zn coordination. The subtracted H atom is bound to the neighbouring O atom. The cleavage of the methylene C−H bond in n-$C_3H_7$ proceeds homolytically and requires 1.22 eV. The apparent barrier of the whole process is 1.81 eV.

For comparison, we also calculated PDH on single-site $ZnO_x$/S-1, created upon replacing one T5-site Si atom (Extended Data Fig. 5c). In comparison with the binuclear $ZnO_x$ species, the first C−H bond cleavage is favourable on this site, but the whole energy span and the apparent activation barrier are higher, that is, 3.39 versus 2.38 eV and 2.14 versus 1.81 eV, respectively (Fig. 2). Therefore, single-site $ZnO_x$ should be much less active, in agreement with the experimental data (Fig. 1h).

The application potential of our approach for preparation of active, selective and durable catalysts was proven in a test over about 400 h on propane stream at 550 °C using different feeds (40 or 70vol% $C_3H_8$ and $H_2$/$C_3H_8$ of 0, 0.5 or 1) representative for the current large-scale PDH technologies. The test consisted of a series of PDH, regeneration (air) and reduction (50vol% $H_2$ in $N_2$) cycles lasting for about 4–6, 0.5–1 and 0.5–1 h respectively at the same temperature. A mixture of ZnO and S-1_3 with an upstream-located ZnO layer (ZnO//ZnO−S-1_3) (Extended Data Fig. 7a) and an analogue of commercial K−$CrO_x$/$Al_2O_3$ were loaded into two reactors of the same setup and tested in parallel. To compare these catalysts fairly, we adjusted contact time to achieve similar initial propane conversion under each reaction condition.

Both catalysts restore their initial activity after regeneration (Fig. 3a). A decrease in the propane conversion over ZnO//ZnO−S-1_3 after about 250 h on propane stream is due to consumption of ZnO under reductive conditions. However, the initial conversion is completely recovered after fresh ZnO was added on top of the remaining catalyst. No visible changes in the conversion could be seen for the next 120 h on

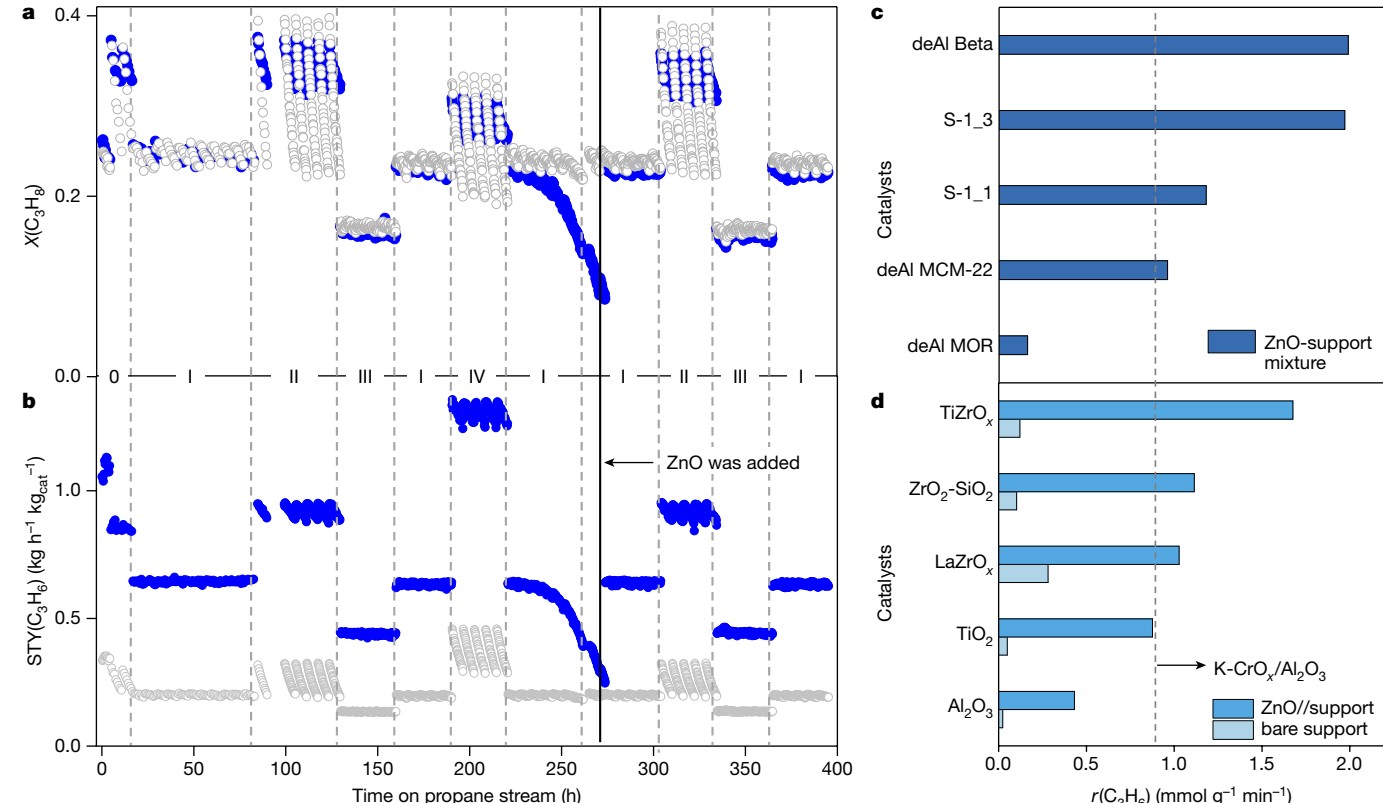

**Fig. 3 | Durability test and validation of the approach. a, b,** On-stream profiles of $X(C_3H_8)$ (**a**) and $STY(C_3H_6)$ (**b**) over ZnO//ZnO−S-1_3 (blue, 0.09 g of ZnO and 0.09 g of ZnO−S-1_3) and K−CrO$_x$/Al$_2$O$_3$ (grey, 0.302 g). Only the amount of ZnO−S-1_3 was considered for calculating $STY(C_3H_6)$. Reaction conditions: 550 °C, (0): $C_3H_8$:$H_2$:$N_2$ = 4:2:4, 10 or 6 ml min$^{-1}$, (I):

$C_3H_8$:$H_2$:$N_2$ = 4:2:4, 6 ml min$^{-1}$, (II) $C_3H_8$:$N_2$ = 4:6, 6 ml min$^{-1}$, (III) $C_3H_8$:$H_2$:$N_2$ = 4:4:2, 6 ml min$^{-1}$, and (IV) $C_3H_8$:$N_2$ = 7:3, 6 ml min$^{-1}$. **c, d,** $r(C_3H_6)$ over ZnO−zeolite mixtures (reduced for 1 h) (**c**) or ZnO//metal oxides (reduced for 2 h) (**d**). A scheme of reactor loading for **c, d** is present in Extended Data Fig. 7e and 7a, respectively.

propane stream. Such behaviour proves that OH nests of the support are thermally stable at the reaction temperature and can easily form catalytically active ZnO$_x$ species. Thus, non-interrupted PDH using our layered catalyst can be ensured on the large-scale application through periodic addition of ZnO, for example, upon non-toxic and explosive-free catalyst regeneration in air.

In comparison with K−CrO$_x$/Al$_2$O$_3$, ZnO//ZnO−S-1_3 reveals higher on-stream stability particularly under conditions without co-fed H$_2$. It also shows about three times higher space time yield of propene formation ($STY(C_3H_6)$) (Fig. 3b) with a slightly lower propene selectivity (on average 91.9% versus 93.1%) due to a higher coke production (Extended Data Fig. 7b, c). However, our catalyst produces a lower amount of cracking products at even 600 °C that might be advantageous for downstream distillation processes (Extended Data Fig. 7d, f). Concerning undesired coke production, this product in the Catofin process is combusted upon catalyst regeneration, thus providing heat required for the endothermic PDH reaction.

Further we demonstrate that our approach is of general character and can be applied for preparation of active Zn-containing catalysts using other SiO$_2$-based zeolites, that is, dealuminated mordenite, Beta and MCM-22 or commercially available OH-rich metal oxides, such as Al$_2$O$_3$, TiO$_2$, LaZrO$_x$, TiZrO$_x$ and ZrO$_2$−SiO$_2$. When applying these supports as a physical mixture with ZnO or in form of two layers with ZnO being the upstream layer, high $r(C_3H_6)$ is achieved (Fig. 3c, d). The activity of most of our samples is higher than that of an analogue of commercial K−CrO$_x$/Al$_2$O$_3$. The mechanism of formation of ZnO$_x$ species in the zeolites and the oxidic supports should be same because H$_2$ formation was observed upon heating of a physical mixture of metallic Zn and support in Ar and the intensity of band of defective

OH groups decreases when such mixtures were treated at 550 °C in H$_2$/N$_2$ = 1 (Extended Data Fig. 8).

Our results demonstrate that defective OH sites in zeolites or metal oxides are crucial for the formation of certain ZnO$_x$ species, which are relevant for the non-oxidative dehydrogenation of propane to propene. Although the concentration of such defects can be adjusted in a controlled manner through the method of support preparation, further improvements are expected when their exact structure and its impact on oxidation of Zn$^0$ to ZnO$_x$ of a certain speciation will be understood. Thus, this study opens the possibility for a purposeful creation of active species and for establishing fundamentals relevant for efficient C−H bond activation in various alkanes. In addition, the catalysts prepared according to our approach might find their application in other Zn-catalysed reactions.

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

# Methods

## Synthesis of bare supports

The following chemicals were used as received. Tetraethyl orthosilicate (Sinopharm Chemical Reagent Co. Ltd or Alfa Aesar), tetrapropylammonium hydroxide (TPAOH, Shanghai Cairui Chemical Engineering Technology Co., Ltd, 25wt% water solution), cetyltrimethylammonium bromide (CTAB, Aldrich), P123 (MW = 5,800, Aldrich), ZnO (Sigma-Aldrich), metallic Zn (>98%, <65 μm, ROTH), tetrabutylammonium hydroxide solution (TBAOH, 40wt%, Sigma-Aldrich), LUDOX (Sigma-Aldrich, 30wt% suspension in $H_2O$), $SiO_2$ (Sigma-Aldrich, with 6 nm pores, 302 $m^2 g^{-1} S_{BET}$ (BET surface area)), MOR ($SiO_2/Al_2O_3 = 20$), MCM-22 ($SiO_2/Al_2O_3 = 46$), beta zeolite ($SiO_2/Al_2O_3 = 25$,) were purchased from NanKai university. γ-$Al_2O_3$ (SAINT-Gobain), $TiO_2$ (rutile $TiO_2$, Sachtleben Chemie GmbH). $TiZrO_x$ (30wt% $TiO_2$) and $LaZrO_x$ (9wt% $La_2O_3$) were offered by Daiichi Kigenso Kagaku Kogyo Co.

S-1_1 was exactly synthesized according to our previous study with an initial gel molar composition of 0.24 TEOS: 0.042 TPAOH: 3.67 $H_2O$ (ref. [31]). S-1 supports with different concentration of OH nests were synthesized with the same procedure upon varying $H_2O/SiO_2$ ratio, silica source, $OH^-/H_2O$ ratio, crystallization temperature or adding $Na^+$. Further details are given in Extended Data Table 1.

To prepare S-2 ($S_{BET}$ of 362 $m^2 g^{-1}$), 27 g of silica sol (30wt%), 18 g of TBAOH solution and 40 g of deionized water were blended at room temperature for 30 min. Hereafter, the suspension was placed in a 200 ml stainless-steel autoclave with a PTFE insert at 170 °C for three days. After cooling down, the solid product was collected after filtration, washing, drying and calcination at 550 °C for 5 h. To remove any $K^+$ and/or $Na^+$, the S-2 zeolite was dispersed in 1 M $NH_4NO_3$ solution ($m_{(solid)}: V_{(solution)} = 1$ g: 50 ml) at 100 °C for 10 h. After ion-exchange, the solid product was washed three times by deionized water and dried at 100 °C for 6 h. The final sample was obtained after calcination at 550 °C for 5 h.

For preparation of SBA-15 (7 nm pores, $S_{BET}$ of 812 $m^2 g^{-1}$), 600 g of deionized water and 120 g of concentrated HCl (37wt%) were mixed at room temperature. Then, 24 g of P123 were added to the above solution followed by adding 48 g of TEOS. The clear solution was heated under stirring to 45 °C and tempered for 30 min. The solution changed to white suspension. This suspension was stirred at 45 °C for additional 24 h and then aged at 90 °C for 48 h under static conditions. After cooling down, the product was separated by filtration and washed thoroughly with deionized water. The sample was dried at room temperature for 24 h and then calcined in air at 550 °C for 6 h with a heating rate of 5 °C $min^{-1}$.

MCM-41 (3 nm pores, $S_{BET}$ of 925 $m^2 g^{-1}$) was synthesized as follows. 500 g of deionized water, 200 g of ethanol and 17 g of CTAB were mixed at room temperature. Then, 42 g of TEOS were added to the above solution. To get an acidic solution (pH value of 2), 0.4 ml of concentrated HCl was added. The solution become clear after a short time. After 30 min, the pH value was adjusted to 8–9 through adding concentrated $NH_4OH$ (25wt%). The white solid product obtained by filtration was washed with deionized water. After drying at room temperature for 24 h, the sample was calcined at 550 °C for 5 h.

To obtain aluminium-free zeolites, 10 g of MOR, Beta or MCM-22 were treated in 200 ml of concentrated $HNO_3$ at 120 °C for 10 h. The washed and dried materials were dispersed and stirred in diluted $HNO_3$ (200 ml, $V_{(HNO3)}: V_{(H2O)} = 3:1$) at room temperature for 5 h to remove extra-framework Al species. Afterwards, the dealuminated zeolites were washed thoroughly by deionized water until pH value reached 7. The washed samples were dried at 100 °C overnight and used without calcination.

## Catalyst preparation

Physical mixtures of ZnO and a certain support were prepared as follows. 0.08 g of ZnO powder and 0.92 of $SiO_2$-based supports were ground in a mortar for 10 min. Afterwards, the resulting mixtures were pressed and sieved to 315–710 μm. The samples were denoted as ZnO support.

A simple method was developed for depositing $ZnO_x$ species on the surface of various supports (inset in Fig. 1b). 0.05 g of ZnO (315–710 μm) and 0.1 g of support (315–710 μm) were loaded into continuous-flow fixed-bed quartz reactors (dimeter: 6 mm), ZnO was located upstream. These two layers were separated by a layer of quartz wool (-10 mg) to avoid any physical contact between ZnO and the supports. The loaded reactors were heated up to 550 °C in Ar flow followed by feeding air flow at the same temperature for 1 h. Hereafter, the catalyst precursors were exposed to a flow (10 ml $min^{-1}$) of 50vol% $H_2$ in Ar, 5vol% $O_2$ in Ar, or Ar at the same temperature and kept for different time (from 5 to 240 min). Finally, they were cooled in Ar. The downstream layer was recovered from the reactors and the resulting catalysts were abbreviated as support(atmosphere). For example, S-1_1($H_2$) means that S-1_1 support was used and synthesized under $H_2$. According to inductively coupled plasma atomic emission spectroscopy measurements, Zn loading in S-1_1($H_2$), MCM-41($H_2$) and S-2($H_2$) is 1.2wt%, 0.98wt% and 2.4wt%, respectively.

An analogue of commercial $K-CrO_x/Al_2O_3$ was synthesized according to the method described in the patent from Vladimir Fridman[32]. Briefly, desired amounts of $CrO_3$ and KOH were separately dissolved in water. Then, both solutions were mixed. Afterwards, γ-$Al_2O_3$ was added to the above solution. The catalyst was collected after drying and calcination at 760 °C for 4 h. The amount of $Cr_2O_3$ and $K_2O$ in the resulting catalyst was 19.7wt% and 0.93wt%, respectively.

## Catalyst characterization

X-ray absorption near-edge structure (XANES) and extended X-ray absorption fine structure (EXAFS) spectra at the Zn K absorption edge were recorded at the P65 beamline of the PETRA III synchrotron (DESY, Hamburg) in transmission mode. The energy of the X-ray photons was selected by a Si(111) double-crystal monochromator and the beam size was set by means of slits to 0.2 (vertical) × 1.5 (horizontal) $mm^2$. The spectra were normalized and the EXAFS spectra background was subtracted using the ATHENA program from the IFFEFIT software package[33]. The $k^2$-weighted EXAFS functions were Fourier transformed (FT) in the k range of 2.5–12.3 $Å^{-1}$ (2.0–10.0 $Å^{-1}$ for the in situ data). Then the amplitude reduction factor $S_0^2 = 1.08$ was obtained by fitting the ZnO reference spectrum to a wurtzite structural model as reported in the Inorganic Crystal Structure Database (ICSD; collection code is 34477). The fits of the EXAFS data were performed using Artemis[33] by a least square method in r space between 1.0 and 3.2 Å. The model with two shells from the wurtzite structure (Zn–O and Zn–Zn or Zn–Si) was used for the fits. Coordination numbers (CN), interatomic distances (r), energy shift (δ$E_0$) and mean square deviation of interatomic distances ($σ^2$) were refined during fitting. The absolute misfit between theory and experiment is represented by $ρ$.

For ex situ catalyst characterisation, samples were diluted with cellulose and the total mass was calculated in such a way that absorbance was 2.5 absorbance units (Lambert–Beer law, the logarithm of the ratio of incident to transmitted (X-ray) photons through a sample). For in situ XANES, the S-1_1($H_2$) catalyst with a sieve fraction of 100–200 μm was loaded in an in situ micro-reactor (quartz capillary, 1.5 mm diameter, 0.02 mm wall thickness[34]). The sample was heated stepwise to 500 °C under 5vol% $H_2$ in He (30 ml $min^{-1}$ flow rate). Before recording the spectra, the sample was kept at each temperature for 10 min.

In situ diffuse reflectance Fourier-transform infrared spectroscopy (DRIFTS) measurements were performed on a Bruker VERTEX 70 equipped with a ZnSe window at a resolution of 4 $cm^{-1}$ from 600 $cm^{-1}$ to 4,000 $cm^{-1}$. Before recording spectra, the background spectrum was obtained over KBr powder at 550 °C in $H_2/Ar = 1$. For in situ DRIFTS measurements, bare support or its physical mixture with ZnO (10wt%) was heated to 550 °C in $N_2$ and then exposed to $H_2/Ar = 1$ for 90 min.

The spectra were recorded every 5 min automatically. For comparative purposes, all spectra were normalized by the intensity of the overtone band of T-O-T lattice vibration at 1,870 cm$^{-1}$ (ref. [28]).

In situ electron paramagnetic resonance (EPR) tests were performed on Bruker ELEXSYS 500-10/12 X-band cw-spectrometer with a microwave power of 6.3 mW, a modulation frequency of 100 kHz and modulation amplitude of 5 G. About 30 mg of catalyst were loaded into an in house-developed continuous-flow fixed-bed quartz reactor, and heated to 550 °C in flow of Ar. Then the sample was exposed to a flow of H$_2$/Ar = 1 (20 ml min$^{-1}$) for 1 h at 550 °C. The EPR spectra were recorded at 20 °C before and after H$_2$ reduction.

Temperature-programmed measurements of physically mixed metallic Zn$^0$ and support were carried out in an in-house developed setup containing eight individually heated continuous-flow fixed-bed quartz reactors. 100 mg of metallic Zn$^0$ was mixed with 200 mg of support. The obtained mixture was then pressed and sieved to 315–710 μm. 50 mg of this fraction was loaded into a quartz tube and flushed with Ar at 40 °C for 90 min. The samples were heated in Ar from 40 to 400 °C (the melting point of Zn is about 420 °C) with a heating rate of 10 °C min$^{-1}$ while recording H$_2$O ($m/z$ = 18) and H$_2$ ($m/z$ = 2) signals by an on-line mass spectrometer.

An overall concentration of OH groups in differently prepared S-1 zeolites (Extended Data Table 1) was determined from the amount of water released upon temperature-programmed catalyst treatment in Ar flow according to our previous work[35]. Each sample (50 mg) was loaded into a quartz tubular reactor and heated up to 550 °C in Ar and then calcined under air at same temperature for 1 h. Then, the samples were reduced in H$_2$/Ar = 1 at 550 °C for 1 h. After that, the samples were cooled down to 400 °C to get stable mass spectrometric signals of gas-phase components. Desorption profiles of water were obtained during heating up to 900 °C with a heating rate of 10 °C min$^{-1}$ in Ar and keeping at the final temperature for 75 min. The amount of released water was determined from the obtained water profiles through their integration over time.

## Temporal analysis of products

Transient studies were performed in the temporal analysis of products (TAP-2) reactor, a time-resolved technique with a resolution of around 100 μs (refs. [30,36]). Each catalyst sample ($m$ = 27 mg, sieve fraction of 315–710 μm) was packed between two layers of quartz particles (sieve fraction of 250–350 μm) within the isothermal zone of the micro-reactor made of quartz. Prior to the pulse experiments, the catalyst was treated in a flow of H$_2$ (5 ml min$^{-1}$) at 550 °C for 0.5 h. Then, it was exposed to high vacuum of about 10$^{-5}$ Pa.

Pulse experiments were performed at 550 °C using D$_2$/Ne = 1, C$_3$H$_8$/Ar = 1, C$_3$D$_8$/Ar = 1 and O$_2$/Ar = 1 mixtures. D$_2$ (CK Special Gases Limited, 2.8), C$_3$H$_8$ (Linde, 3.5), C$_3$D$_8$ (ISOTEC INC., 99 at% D), O$_2$ (Air Liquide, 4.8), Ar (Air Liquide, 5.0) and Ne (Air Liquide, 4.0) were used without additional purification. Transient responses at the reactor outlet were monitored with a quadrupole mass spectrometer (HAL RD 301 Hiden Analytical) at atomic mass units (AMU) related to the reactants, reaction products and inert gases (Ar and Ne). The latter were used as reference standards. The following AMUs were recorded 52 (C$_3$D$_8$), 48 (C$_3$D$_6$), 46 (C$_3$H$_6$D$_2$), 45 (C$_3$H$_7$D), 44 (C$_3$H$_7$D, C$_3$H$_8$), 42 (C$_3$H$_8$, C$_3$H$_6$), 41 (C$_3$H$_8$, C$_3$H$_6$), 34(C$_3$D$_8$), 32 (O$_2$), 31 (C$_3$H$_6$D$_2$), 30 (C$_3$H$_7$D), 29 (C$_3$H$_8$), 28 (C$_3$H$_8$), 18.0 (H$_2$O), 4.0 (D$_2$), 3.0 (HD), 2.0 (H$_2$), 40 (Ar) and 20 (Ar, Ne). For each AMU, pulses were repeated ten times and averaged to improve the signal-to-noise ratio. The concentration of the feed components and the reaction products was determined from the respective AMU using standard fragmentation patterns and sensitivity factors determined in separate calibration tests.

## Catalytic tests

PDH tests were carried out in an in-house developed setup with 15 continuous-flow tubular fixed-bed quartz reactors. The total pressure

was 1.2 bar (absolute). For determining the rate of propene formation ($r$(C$_3$H$_6$)), the degree of propane conversion was below 10% when using 50 mg of catalyst and a total flow of 40 ml min$^{-1}$ containing 40vol% C$_3$H$_8$ in N$_2$. Before the catalytic tests, the samples were initially heated up to 550 °C in N$_2$, and then flushed in air at the same temperature for 1 h. After 15 min purging with N$_2$, for oxidatively treated samples, the catalysts were directly exposed to 40vol% C$_3$H$_8$ in N$_2$, while for reductively treated samples, the catalysts were additionally reduced in H$_2$/N$_2$ = 1 for 1 h followed by flushing with N$_2$ and finally feeding 40vol% C$_3$H$_8$ in N$_2$. $r$(C$_3$H$_6$) is calculated according to equation (1).

$$r(C_3H_6) = \frac{\dot{n}(C_3H_6)}{m_{cat}}. \tag{1}$$

where $\dot{n}$(C$_3$H$_6$) and $m_{cat}$ stand for the molar flow of propene (mmol min$^{-1}$) and catalyst mass, respectively.

A durability PDH test was carried out at 550 °C with different feeds for about 400 h on propane stream (total time is about 600 h). A catalyst consisting of a ZnO layer on top of ZnO–S-1_3 (ZnO//ZnO–S-1_3) and an analogue of commercial K–CrO$_x$/Al$_2$O$_3$ were tested in the same setup in parallel. A scheme representing reactor loading in these tests is shown in Extended Data Fig. 7a. To achieve a similar degree of propane conversion, we used 0.302 g of K–CrO$_x$/Al$_2$O$_3$ as well as 0.09 g of ZnO and 0.09 g of ZnO–S-1_3. After about 250 h on propane stream, 0.09 g of ZnO were added on top of S-1_3 because the first ZnO layer was consumed. The test consisted of a series of dehydrogenation, regeneration cycles and reduction. The catalysts were purged with N$_2$ for 15 min between each step.

To determine an apparent activation energy of propene formation, $r$(C$_3$H$_6$) over ZnO–S-1_1, ZnO–S-1_2, ZnO–S-1_3, ZnO–S-1_4 and ZnO–S-1_5 was determined at 500, 525 and 550 °C. The desired values were obtained by plotting ln($r$C$_3$H$_6$) versus 1/$T$ (Extended Data Fig. 1j). Propane conversion was kept below 10% to ensure differential operation.

The conversion of propane and the selectivity to gas-phase products and to coke were calculated according to equations (2), (3) and (4), respectively. Equation (5) was used for calculating the space time yield (STY) of propene formation. The turnover frequency (TOF) of propene formation was calculated according to equation (6).

$$X(C_3H_8) = \frac{\dot{n}^{in}_{C_3H_8} - \dot{n}^{out}_{C_3H_8}}{\dot{n}^{in}_{C_3H_8}} \tag{2}$$

$$S(i) = \frac{\beta_i}{\beta_{C_3H_8}} \frac{\dot{n}^{out}_i}{\dot{n}^{in}_{C_3H_8} - \dot{n}^{out}_{C_3H_8}} \tag{3}$$

$$S(coke) = 1 - \sum_i S(i) \tag{4}$$

$$STY = \frac{\dot{n}_{C_3H_6} \times M_{C_3H_6} \times 60}{1,000 \times m_{cat}} \tag{5}$$

$$TOF = \frac{r(C_3H_6) \times 60}{n_{Zn}} \tag{6}$$

Where $\dot{n}$ with superscripts 'in' or 'out' means the molar flow of gas phase components at the reactor inlet and outlet (mmol min$^{-1}$), respectively. $\beta_i$ represents the number of carbon atoms in propane and the product $i$. M$_{C_3H_6}$ is the molecular weight of propene (42 g mol$^{-1}$). $n_{Zn}$ means the molar loading of Zn (mmol g$^{-1}$).

The feed components and the reaction products were analysed by an on-line gas chromatograph (GC, Agilent 6890) equipped with flame ionization (FID) and thermal conductivity (TCD) detectors. Gas-phase components were identified by GC, which equipped with PLOT/Q (for CO$_2$), AL/S (for hydrocarbons) and Molsieve 5 (for H$_2$, O$_2$, N$_2$ and CO) columns. The GC analysis time of gas-phase products is 4 min.

## DTF calculations

Spin-polarized and periodic density functional theory calculations were carried out with the VASP software[37,38]. The electron exchange and correlation energies were treated using the generalized gradient approximation in the GGA-PBE functional[39,40]. The cut-off energy was set up to 400 eV. Geometry optimization was converged until forces acting on atoms were lower than 0.02 eV Å$^{-1}$, and the energy difference was lower than $10^{-4}$ eV. The CI-NEB method was applied for identifying transition states[41]. All our reported energies include dispersion (D3)[42] and ZPE corrections (PBE + D3 + ZPE). In addition, we tested the corrections of the Hubbard term (DFT + $U_{eff}$)[43] for partially reduced oxidation state of Zn atom ($U_{eff}$ = 4.7 eV (ref. [44])) for comparison. The $U_{eff}$ term just affects the absolute value of states in potential energy surface (PES) and does not change the reaction trend or the mechanism. Therefore, we used the PBE functional results for main discussion and the DFT + $U_{eff}$ results for comparison.

A model with binuclear $ZnO_x$ species inside the pores of S-1 (MFI) was used for calculating elementary pathways of PDH reaction. For comparison, we calculated the PDH reaction on isolated $ZnO_x$ inside the pores of S-1.

## Data availability

All data that led us to understand the results presented here are available with the paper or from the corresponding author upon reasonable request. Source data are provided with this paper.

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

**Acknowledgements** Financial support by Deutsche Forschungsgemeinschaft (KO 2261/8-1, JI 210/1-1), the National Natural Science Foundation of China (grants 21961132026, 21878331, 91645108, 21903049 and U1510103), the National key Research and Development Program Nanotechnology Specific Project (no. 2020YFA0210900), Science Foundation of China University of Petroleum, Beijing (C201604) and the State of Mecklenburg-Vorpommern are gratefully acknowledged. The authors thank K. Guo for synthesis of some zeolites. The authors thank A. Simmula for ICP measurement. The authors thank J. Li for IR measurements of metal oxides. The authors also thank Daiichi Kigenso Kagaku Kogyo Co., Ltd, for offering $ZrO_2$-based supports. D.Z. acknowledges support from the China Scholarship Council. We acknowledge DESY (Hamburg, Germany), a member of the Helmholtz Association HGF, for the provision of experimental facilities. Parts of this research were carried out at PETRA III and we would like to thank E. Welter for assistance in using beamline P65.

**Author contributions** E.V.K. initiated and led the whole project. E.V.K. and G.J. supervised and coordinated the project. D.Z. prepared all the catalysts and carried out characterization measurements and catalytic tests. D.Z. and E.V.K. wrote the first draft. U.R. and D.L. contributed to the analysis of catalytic tests. X.T. and H.J. performed DFT calculations and wrote the corresponding part of the manuscript. D.E.D. and J.D.G. performed XAS experiments and analysed the results. S.H. performed in situ DRIFTS measurements and some catalytic tests. V.A.K. carried out temporal analysis of product tests and analysed the results. T.V. and J.R performed in situ EPR characterization and analysed the results. A.P. carried out the durability test and analysed the results. R.E. synthesized mesoporous $SiO_2$ materials and carried out $N_2$ adsorption–desorption measurements. All the authors discussed the results and improved the manuscript.

**Funding** Open access funding provided by Leibniz-Institut für Katalyse e.V. (LIKAT Rostock).

**Competing interests** The authors declare no competing interests.

**Additional information**
**Correspondence and requests for materials** should be addressed to Guiyuan Jiang, Haijun Jiao or Evgenii V. Kondratenko.

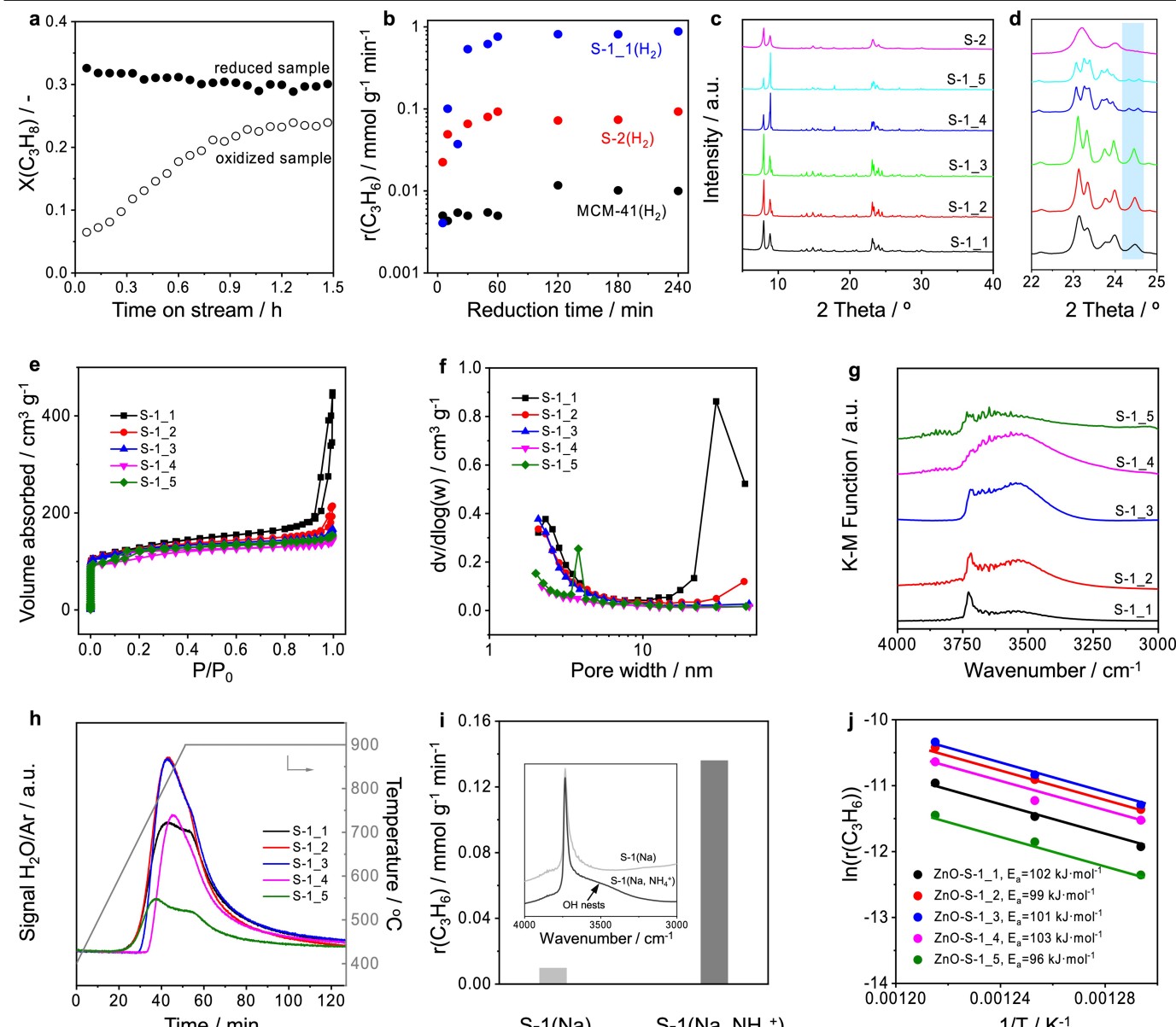

**Extended Data Fig. 1 | Selected catalytic performance and physicochemical characteristics. a**, On-stream profiles of X(C₃H₈) over oxidatively or reductively treated ZnO–S-1_1 mixture (150 mg) at 550 °C using C₃H₈:N₂=2:3 (15 ml min⁻¹). **b**, r(C₃H₆) at 550 °C over S-1_1(H₂), S-2(H₂) and MCM-41(H₂) prepared at 550 °C for different reduction time according to the inset in Fig. 1b. **c**, XRD patterns of differently synthesized S-1 zeolites and S-2 zeolite. **d**, Magnified XRD patterns in the 2θ range of 22–25°. S-1_1, S-1_2 and S-1_3 are composed of the orthorhombic MFI phase, while the monoclinic MFI phase was identified in S-1_4 and S-1_5. The XRD patterns of S-2 are typical for the MEL topology. **e**, N₂ adsorption–

desorption isotherms of S-1 zeolites. **f**, Pore size distribution of S-1 zeolites based on the BJH model. $S_{BET}$ is given in Extended Data Table 1. **g**, DRIFTS spectra of S-1 zeolites in the OH stretching region. **h**, Temperature-programmed release of H₂O upon heating of S-1 zeolites in Ar. **i**, r(C₃H₆) over ZnO–S-1(Na) or ZnO–S-1(Na, NH₄⁺) mixtures. The bare supports differ in the concentration of OH nests (see the DRIFTS spectra in the insert), which are important for the formation of catalytically active ZnOₓ species from Zn⁰. **j**, Arrhenius plots of r(C₃H₆) over ZnO–S-1 mixtures (50 mg) tested using C₃H₈:N₂ = 2:3 (40 ml min⁻¹).

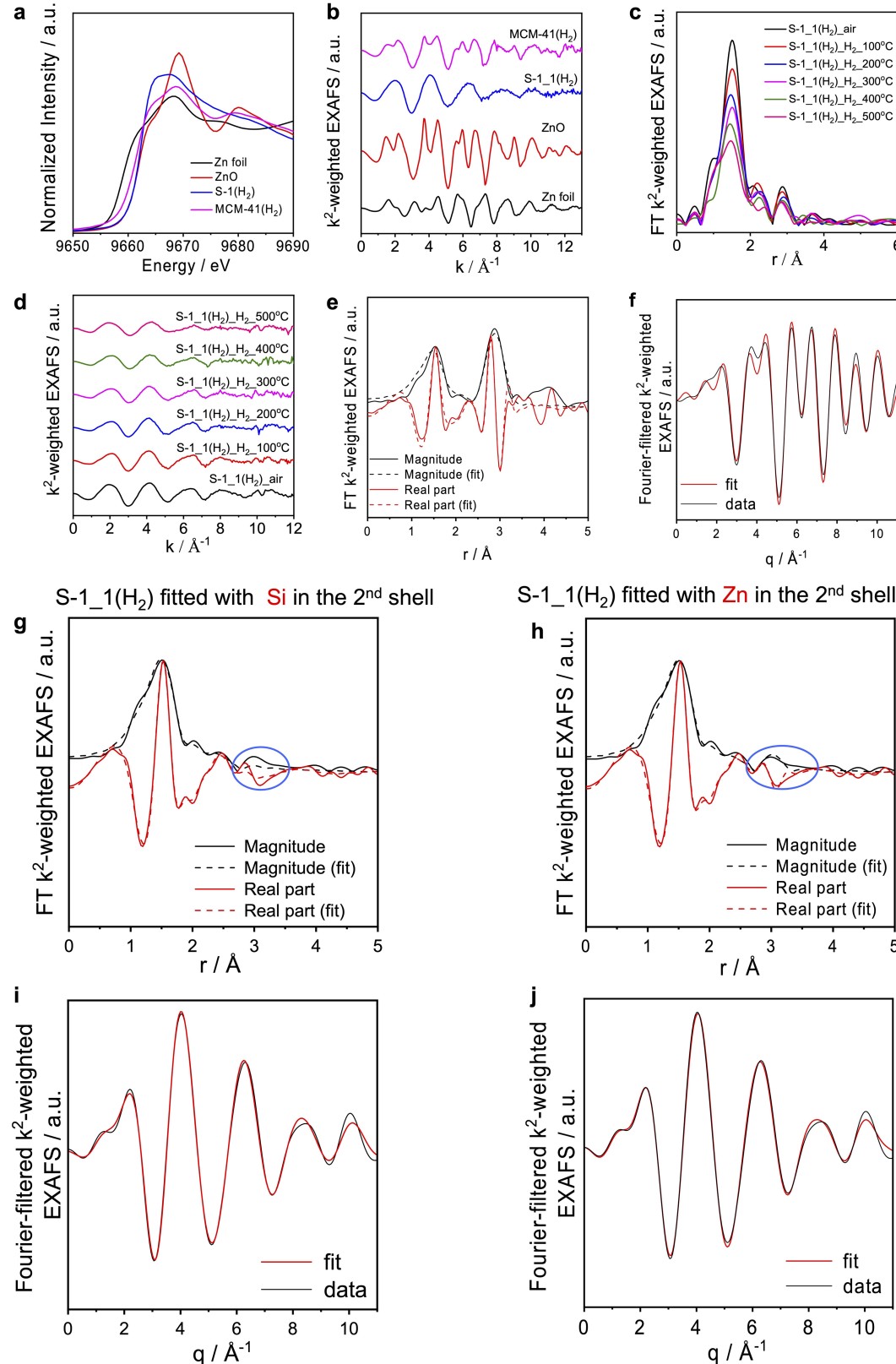

**Extended Data Fig. 2 | Ex situ, in situ XAS and fitting data. a**, XANES spectra of as-prepared catalysts and reference materials (Zn foil and commercial ZnO). **b**, The corresponding $k^2$-weighted $\chi(k)$ functions (extracted fine structure in $k$ space) to **a**. **c**, In situ FT EXAFS spectra of S-1_1(H$_2$) in a flow of 5vol%H$_2$ in He at different temperatures. **d**, The corresponding $k^2$-weighted $\chi(k)$ functions (extracted fine structure in $k$ space) to **c**. **e**, **f**, The EXAFS fits of the ZnO reference (used to estimate amplitude reduction factor) in $r$ (**e**) and q spaces (**f**). ZnO fit summary (based on the wurtzite structure): $\delta E_0 = 1.0 \pm 2.6$ eV; $\rho = 2.0\%$;

amplitude reduction factor ($S_0^2$) = $1.08 \pm 0.2$; $\sigma^2 = 10.1 \pm 1.8$ eV (the same for all fitted shells). First shell comprises four O atoms at $1.96 \pm 0.03$ Å; second shell: one O atom at $3.11 \pm 0.34$ Å, six Zn atoms at $3.21 \pm 0.02$ Å and six Zn atoms at $3.26 \pm 0.02$ Å. **g–j**, Comparison between the ex situ spectrum of S-1_1(H$_2$) and models assuming Si or Zn in the second shell in $r$ (**g**, **h**) and $q$ spaces (**i**, **j**). A significant misfit can be seen for the second shell peak at approx. 3.0 Å for the Si-containing model.

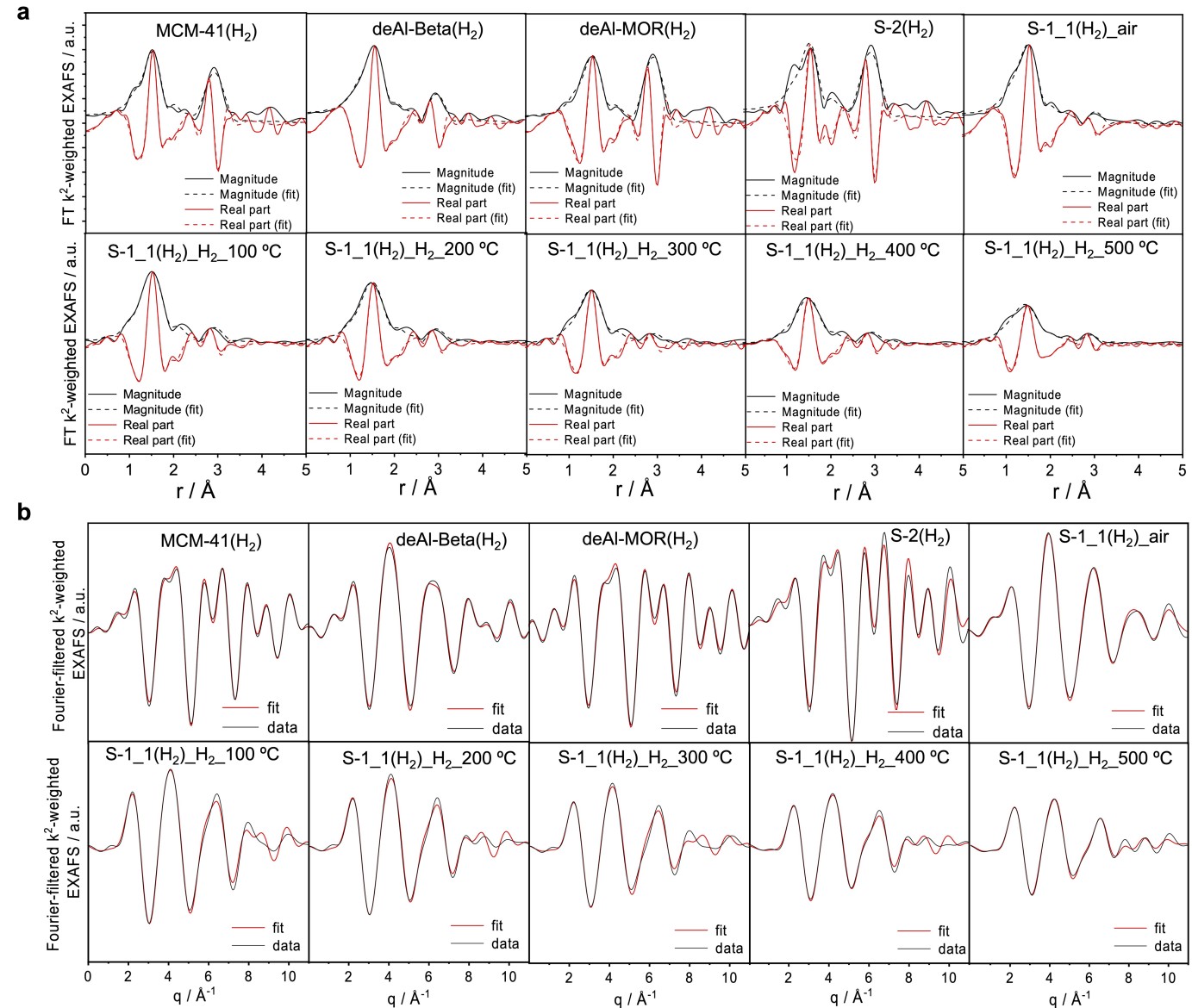

**Extended Data Fig. 3 | Fitting results. a**, **b**, The EXAFS fits for different materials in r (**a**) and *q* spaces (**b**).

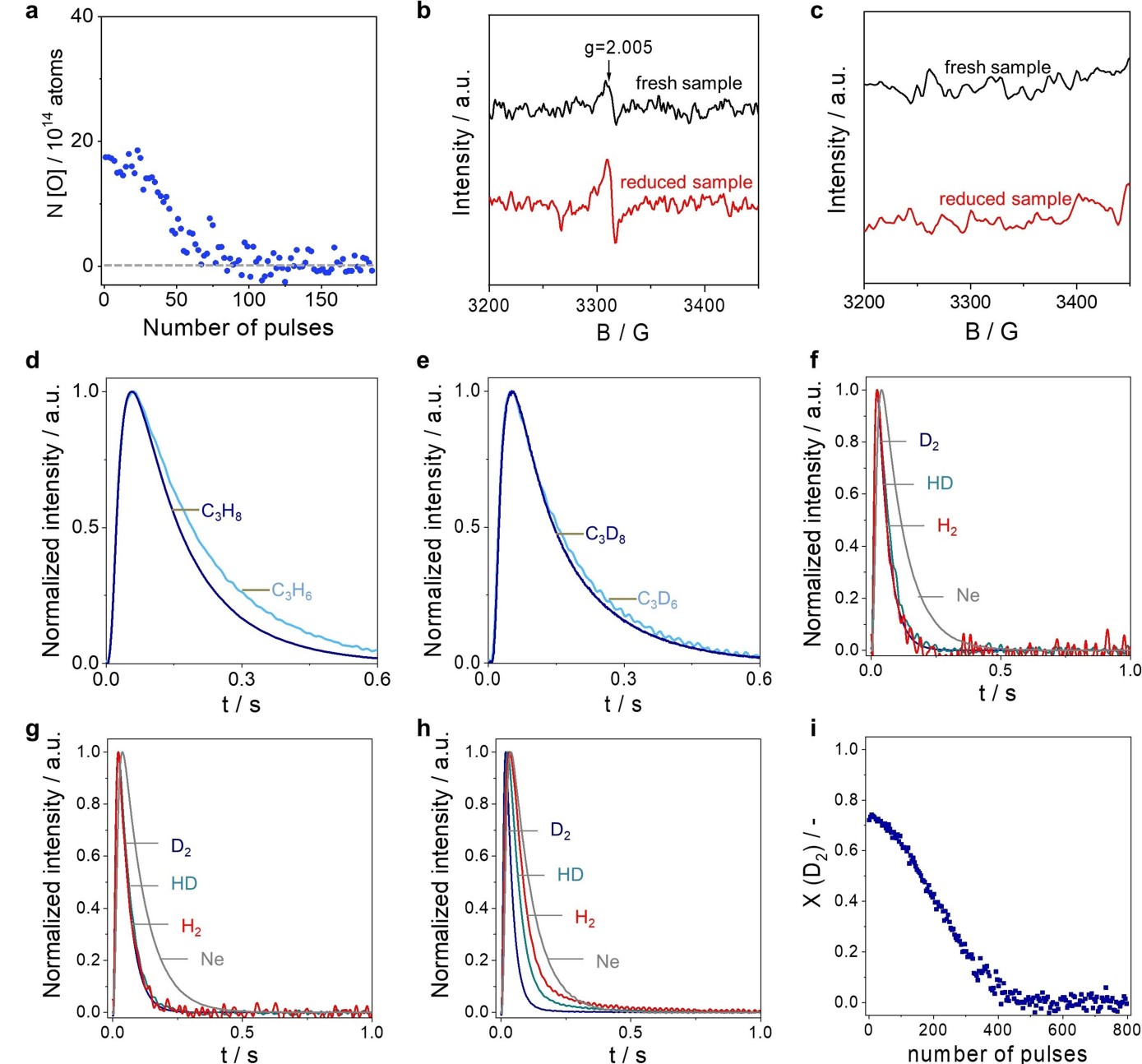

**Extended Data Fig. 4 | Proving the presence of O vacancies and rate-limiting step in PDH. a**, The amount of consumed oxygen atoms (N[O]) upon $O_2$ pulsing at 550 °C over S-1_1($H_2$) reduced in $H_2$/Ar = 1 at 550 °C for 0.5 h. **b**, **c**, In situ EPR spectra at 20 °C of fresh and treated ($H_2$/Ar = 1 (20 ml min⁻¹) at 550 °C for 1 h) S-1_1($H_2$) (**b**) or bare S-1_1 (**c**). **d**, **e**, Height-normalized transient responses of $C_3H_8$, $C_3H_6$, $C_3D_8$, $C_3D_6$, $D_2$ and HD recorded after pulsing of $C_3H_8$/Ar = 1 (**d**) or $C_3D_8$/Ar = 1 (**e**) at 550 °C. **f–h**, Height-normalized transient responses of $H_2$, $D_2$ and H/D recorded after pulsing of $D_2$/Ne = 1 over S-1_1 (**f**), MCM-41($H_2$) (**g**), S-1_1($H_2$) (**h**) at 550 °C. **i**, Conversion of $D_2$ (X($D_2$)) over S-1_1($H_2$) in multi-pulse $D_2$/Ne = 1 tests at 550 °C. Experimental details are given in the 'Temporal analysis of products' section.

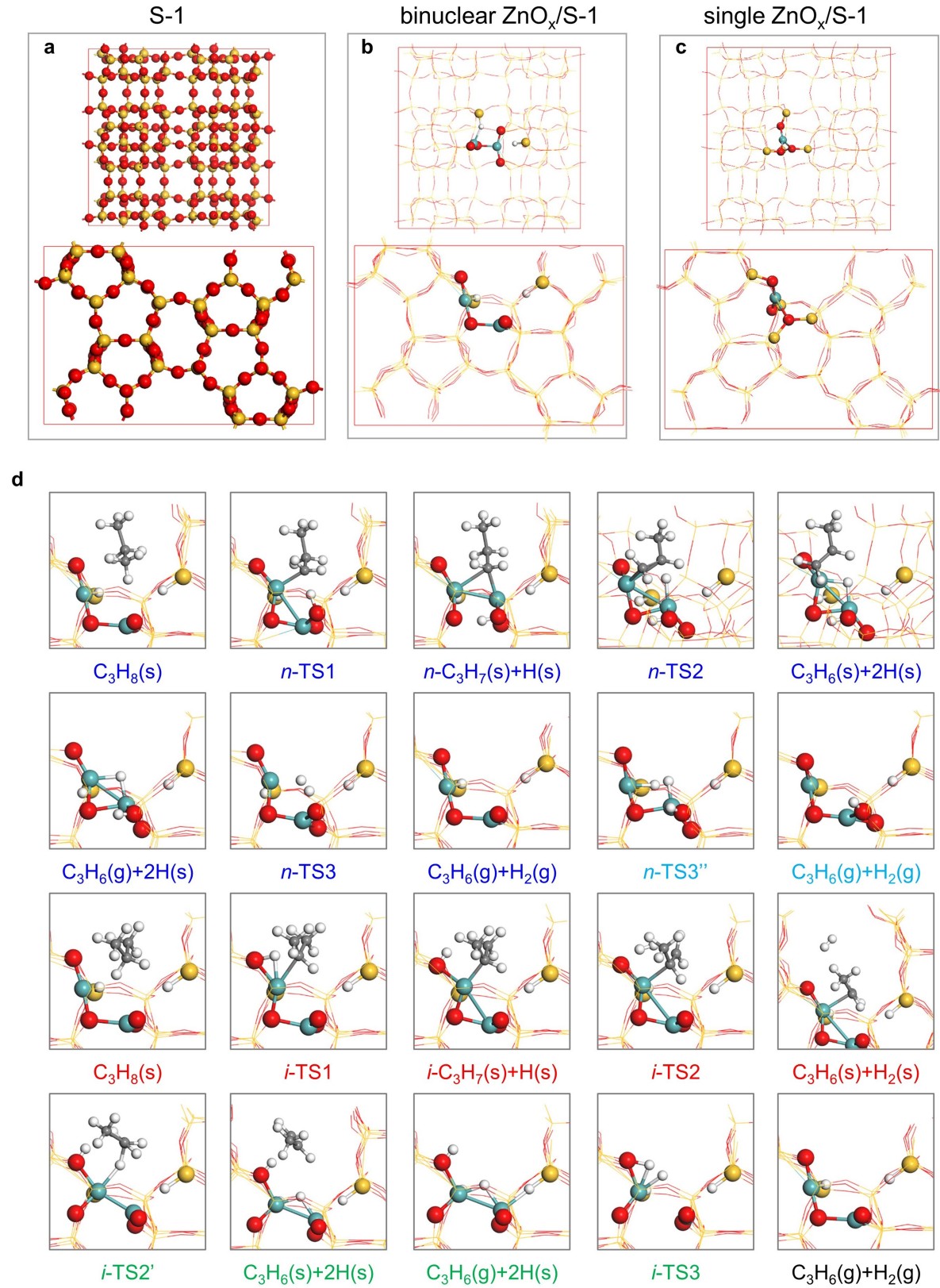

S-1 binuclear ZnO$_x$/S-1 single ZnO$_x$/S-1

**Extended Data Fig. 5** | See next page for caption.

**Extended Data Fig. 5 | Calculation DFT models and intermediates for different reaction pathways. a**–**c**, Top and side views of the MFI cell (S-1 support) (**a**), the binuclear $ZnO_x$/S-1 model (**b**) and the single $ZnO_x$/S-1 model (**c**). **d**, Side views of intermediates in the *n*-propyl or isopropyl paths using the $ZnO_x$/S-1 model. (Cyan, red, grey, yellow and white balls stand for Zn, O, C, Si and H atoms, respectively.) The orthorhombic MFI framework, which contains 12 distinct tetrahedral (T) centres based on a periodic slab model containing 96 T centres was used to simulate (Silicalite-1, S-1) catalyst, especially its channel structure. A $1 \times 1 \times 1$ Monkhorst–Pack *k*-point grid was used for sampling the Brillouin zone for this model. We randomly constructed a defective MFI catalyst, where two neighbouring Si atoms at T1 and T5 sites both located at the intersection of the channels and surrounded by six OH groups forming a OH nest were removed. Then, these two defective sites were filled with two Zn atoms with a simultaneous formation of two $H_2$ molecules, and then further removal the left two Si–OH groups by $H_2$ into two Si–H groups (mimics the additional reduction steps before carrying out the reaction). The OH location, that is, the final Si–H location, is chosen according to the Zn–Zn distance (3.32 Å, EXAFS results).

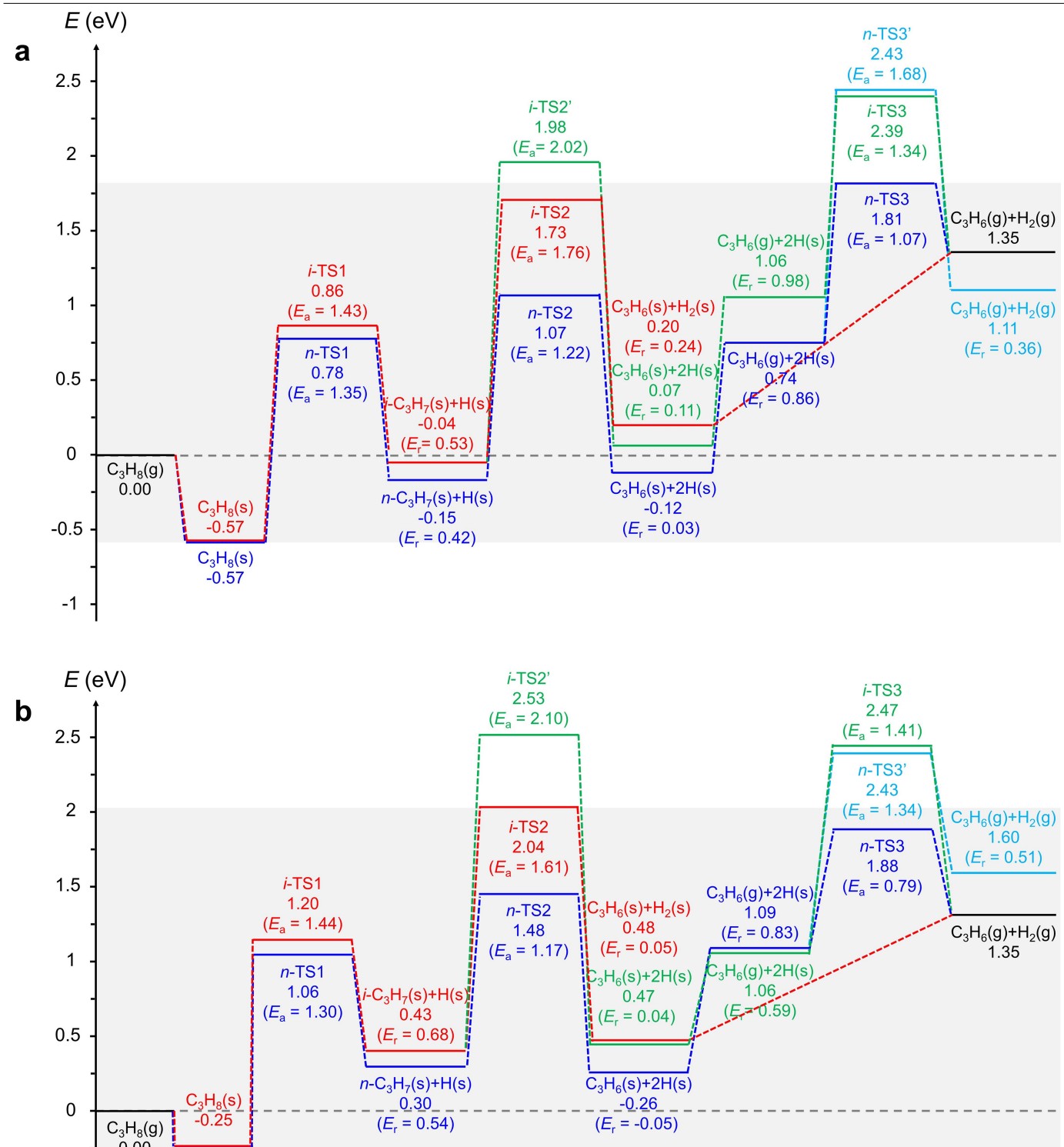

**Extended Data Fig. 6 | Potential energy surface of different reaction pathways in PDH. a**, The potential energy surface of PDH using the binuclear ZnO$_x$/S-1 model at the PBE + D3 + ZPE level. Two competitive PDH paths are considered: (i) propane is initially activated upon methyl C–H heterolytic cleavage to yield *n*-C$_3$H$_7$ or (ii) propane is activated at the central methylene

C–H bond heterolytically to yield *i*-C$_3$H$_7$. The calculations predict that the latter route is unlike. **b**, The $U_{eff}$ corrected potential energy surfaces of PDH using the binuclear ZnO$_x$/S-1 model. The $U_{eff}$ term only affects the absolute value of states in the potential energy surface (PES) and does not change the reaction trend or the mechanism.

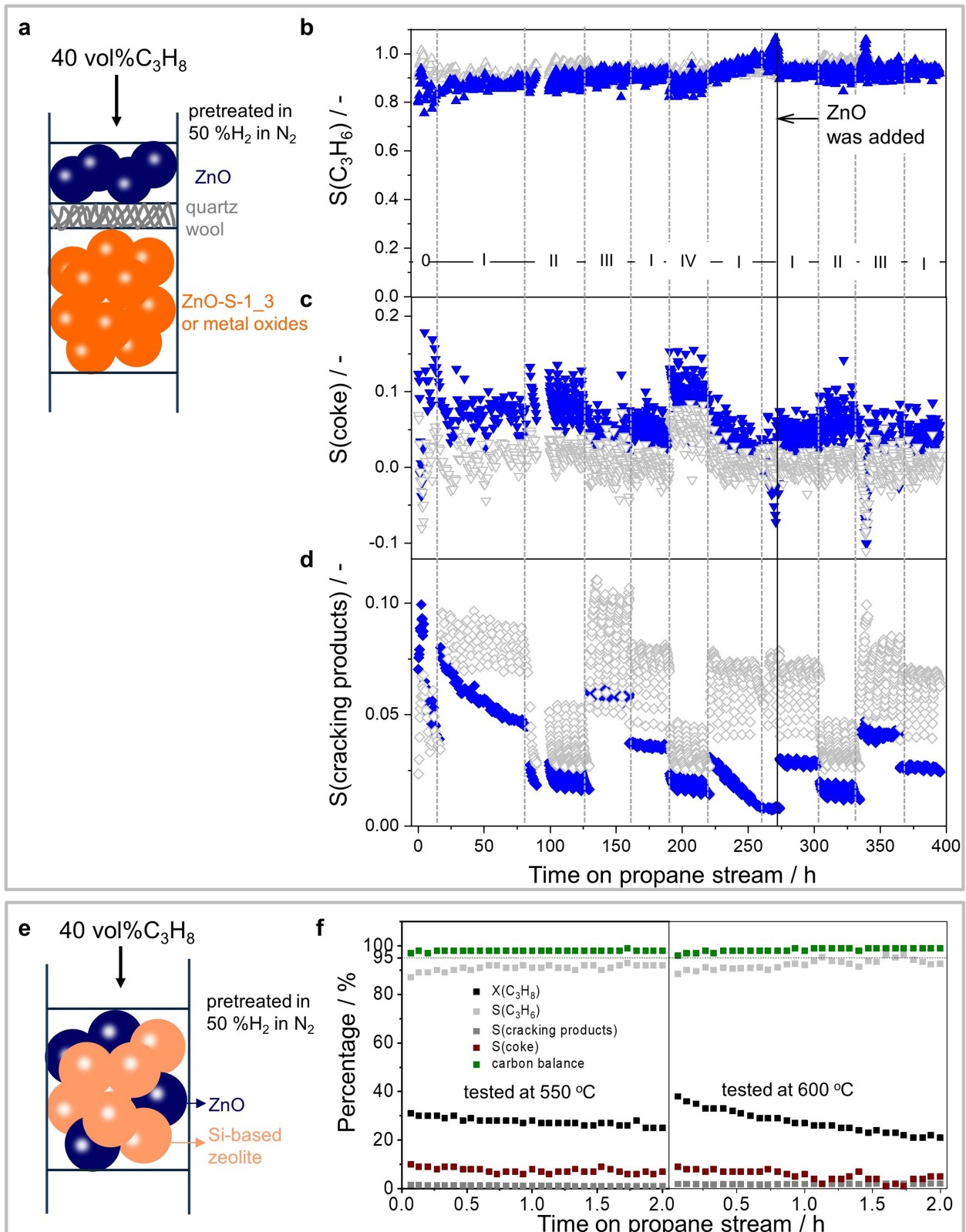

**Extended Data Fig. 7 | Durability of ZnO//ZnO–S-1_3 and K–CrO$_x$/Al$_2$O$_3$ under industrially relevant conditions. a**, A schematic of reactor loading for the tests. **b**–**d**, On-stream profiles of the selectivity to propene (S(C$_3$H$_6$)) (**b**), coke (S(coke)) (**c**) or cracking products (S(cracking products)) (**d**) over ZnO// ZnO–S-1_3 (blue, 0.09 g ZnO and 0.09 g ZnO–S-1_3) and an analogue of commercial K–CrO$_x$/Al$_2$O$_3$ (grey, 0.302 g). Reaction conditions: 550 °C,

(0): C$_3$H$_8$:H$_2$:N$_2$ = 4:2:4, 10 or 6 ml min$^{-1}$, (I): C$_3$H$_8$:H$_2$:N$_2$ = 4:2:4, 6 ml min$^{-1}$, (II) C$_3$H$_8$:N$_2$ = 4:6, 6 ml min$^{-1}$, (III) C$_3$H$_8$:H$_2$:N$_2$=4:4:2, 6 ml min$^{-1}$, and (IV) C$_3$H$_8$:N$_2$=7:3, 6 ml min$^{-1}$. **e**, A schematic of catalytic tests for **f**. **f**, The X(C$_3$H$_8$), S(C$_3$H$_6$), S(cracking products), and S(coke) as well as carbon balance in 2 h propane on stream over ZnO–S-1_3 tested at 550 °C or 600 °C using C$_3$H$_8$:N$_2$ = 2:3 with WHSV$_{(C3H8)}$ of 7.9 or 15.8 h$^{-1}$. All carbon balance values are above 95%.

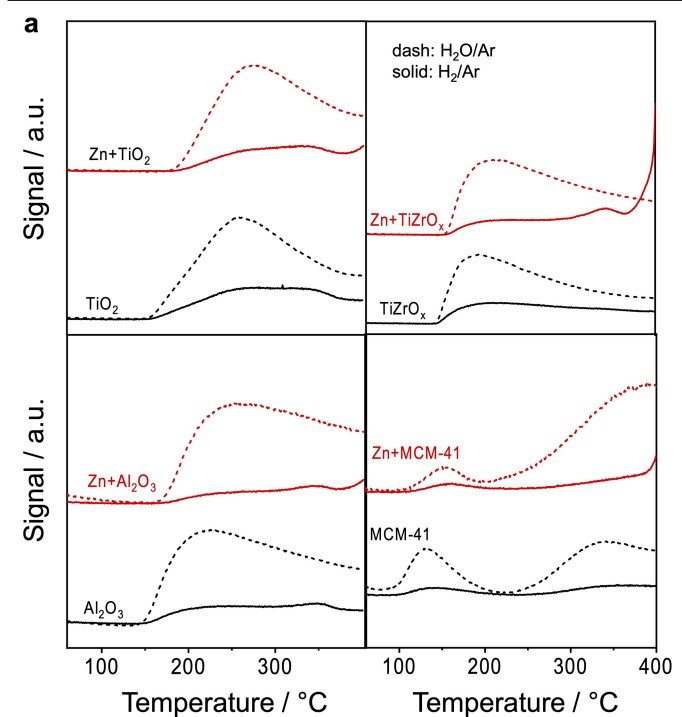

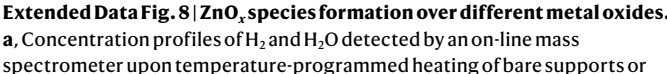

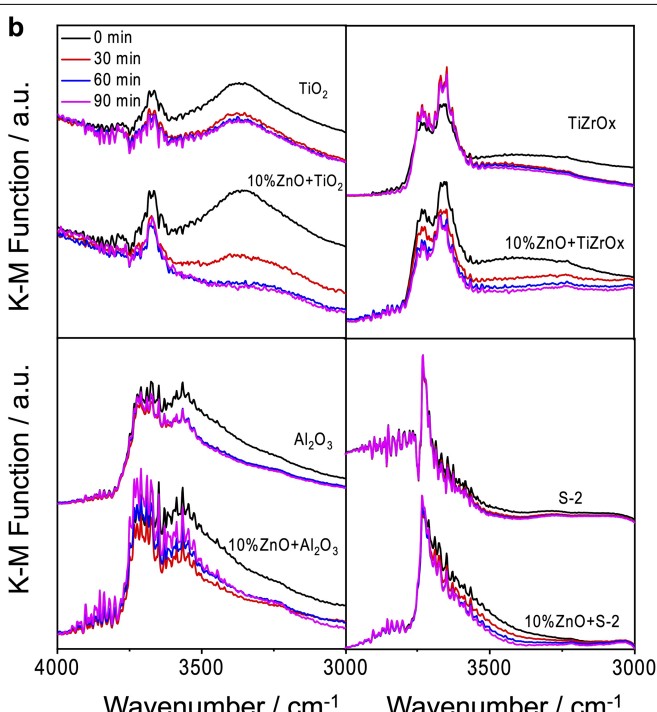

**Extended Data Fig. 8 | ZnO$_x$ species formation over different metal oxides. a**, Concentration profiles of H$_2$ and H$_2$O detected by an on-line mass spectrometer upon temperature-programmed heating of bare supports or physically mixed metallic Zn$^0$ and a certain support in Ar. **b**, In situ DRIFTS spectra of bare supports or a mixture of ZnO with a certain support. Reaction conditions: 550 °C, 50vol%H$_2$ in Ar.

**Extended Data Table 1 | Synthesis conditions and physicochemical properties of differently prepared S-1 supports**

| names | composition of initial gel | conditions | $S_{(BET)}$ / $m^2 g^{-1}$ | $S_{(internal)}$ / $m^2 g^{-1}$ | isolated OH / $cm^{-1}$ | OH nests/ $cm^{-1}$ | ratio $I_{3450}/I_{3720}$ | n(OH nest)/ mmol $g^{-1}$ |
|---|---|---|---|---|---|---|---|---|
| S-1_1 | $0.24TEOS: 0.042TPAOH: 3.7H_2O$ | 100 °C, 48 h | 462 | 376 | 3726 | 3536 | 0.35 | 0.374 |
| S-1_2 | $0.14TEOS: 0.036TPAOH: 3.7H_2O$ | 150 °C, 48 h | 458 | 425 | 3718 | 3535 | 0.82 | 0.764 |
| S-1_3 | $0.24TEOS: 0.036TPAOH: 3.7 H_2O, 0.5g$ seed | 170 °C, 48 h | 452 | 429 | 3716 | 3546 | 1.25 | 0.960 |
| S-1_4 | $0.12TEOS: 0.018TPAOH: 5.3 H_2O$ | 170 °C, 48 h | 395 | 376 | 3718 | 3543 | 1.78 | 0.724 |
| S-1_5 | $0.12$ silica sol: $0.018TPAOH: 5.9 H_2O$ | 170 °C, 48 h | 406 | 383 | 3722 | 3524 | 0.89 | 0.265 |
| S-1(Na) | $0.24TEOS: 0.06TPAOH: 3.5 H_2O:0.012NaOH$ | 100 °C, 48 h | - | - | - | - | - | - |
| S-1(Na, $NH_4^+$) | $0.24TEOS: 0.06TPAOH: 3.5 H_2O:0.012NaOH$ | 80 °C, 1M $NH_4NO_3$, 5 h | - | - | - | - | - | - |

Details of synthesis of different S-1 supports, their specific surface area as well as IR bands characteristic of isolated and OH nests, the ratio of their intensities ($I_{3450}/I_{3720}$) and the overall amount (n(OH nests)) of OH groups in these materials.

Note: '-' means was not measured.

**Extended Data Table 2 | Fitting results**

| names | Zn-O distance (Å) | CN (Zn-O) | $\sigma^2$ for O $(10^{-3} Å^2)$ | Zn-Zn distance (Å) | CN (Zn-Zn) | $\sigma^2$ for Zn $(10^{-3} Å^2)$ | $\delta E_0$ (eV) | $\rho$ (%) |
|---|---|---|---|---|---|---|---|---|
| MCM-41(H$_2$) | 1.95±0.01 | 2.5±0.2 | 7.3* | 3.25±0.01 | 8.4±2.4 | 14.8±3.4 | 2.6±1.1 | 0.9 |
| deAl-Beta(H$_2$) | 1.98±0.01 | 2.9±0.1 | 7.3* | 3.30±0.02 | 6.1±2.1 | 18.7±4.1 | 3.9±0.6 | 0.3 |
| deAl-MOR(H$_2$) | 1.97±0.01 | 2.4±0.2 | 7.3* | 3.25±0.01 | 12.6±2.5 | 16.3±2.0 | 3.3±0.8 | 0.5 |
| S-2(H$_2$) | 1.95±0.03 | 2.4±0.5 | 7.3* | 3.22±0.02 | 12.4±6.2 | 15.3±5.0 | 2.1±1.8 | 2.8 |
| S-1_1(H$_2$) | 1.97±0.01 | 2.9±0.2 | 7.3±1.6 | 3.32±0.07 | 1.3±0.5 | 7.3±1.6 | 4.7±0.6 | 0.2 |
| S-1_1(H$_2$)_H$_2$_ 100 °C | 1.98±0.01 | 2.5±0.2 | 7.2±2.0 | 3.32±0.03 | 1.2±0.5 | 7.2±2.0 | 4.8±0.8 | 0.6 |
| S-1_1(H$_2$)_H$_2$_ 200 °C | 1.97±0.01 | 2.4±0.2 | 9.1±2.1 | 3.34±0.03 | 1.3±0.5 | 9.1±2.1 | 4.4±0.8 | 0.6 |
| S-1_1(H$_2$)_H$_2$_ 300 °C | 1.96±0.01 | 2.1±0.2 | 9.1±2.5 | 3.32±0.04 | 0.9±0.5 | 9.1±2.5 | 4.6±0.9 | 1.0 |
| S-1_1(H$_2$)_H$_2$_ 400 °C | 1.95±0.01 | 1.8±0.2 | 10.0±2.3 | 3.31±0.02 | 1.2±0.5 | 10.0±2.3 | 5.0±0.8 | 0.9 |
| S-1_1(H$_2$)_H$_2$_ 500 °C | 1.92±0.01 | 2.0±0.2 | 14.4±2.4 | 3.30±0.03 | 1.4±0.5 | 14.4±2.4 | 2.6±0.8 | 0.7 |

| | Zn-O distance (Å) | CN (Zn-O) | $\sigma^2$ for O $(10^{-3} Å^2)$ | Zn-Si distance (Å) | CN (Zn-Si) | | $\delta E_0$ (eV) | $\rho$ (%) |
|---|---|---|---|---|---|---|---|---|
| *Ex-situ* EXAFS fit parameters of S-1_1(H$_2$) with Si in the second shell | | | | | | | | |
| S-1_1(H$_2$) fitted with Si in the 2$^{nd}$ shell | 1.95±0.01 | 3.2±0.2 | 7.5±1.2 | 3.12±0.07 | 0.4±0.3 | | 2.1±0.9 | 0.2 |

Ex situ and in situ EXAFS fit parameters of as-synthesized ZnO/Si-based catalysts and S-1_1(H$_2$), respectively.

*Fixed at the most common value for room temperature measurements for the ease of data comparison.

Note: coordination numbers and distances in reference materials:

Zn metal: $CN_1$(Zn–Zn)=6, $r_1$(Zn–Zn)=2.665 Å; $CN_2$(Zn–Zn)=6, $r_2$(Zn–Zn)=2.913 Å.

Bulk ZnO: CN(Zn–O)=4, $r$(Zn–O)=1.970 Å, CN(Zn–Zn)=6, $r$(Zn–Zn)=3.213 Å, CN(Zn–Zn)=6, $r$(Zn–Zn)=3.250 Å.

**Extended Data Table 3 | Comparison of Zn-containing catalysts**

| names | preparation method | WHSV($C_3H_8$) / $h^{-1}$ | conditions | X($C_3H_8$) / - | S($C_3H_6$) / - | STY / kg $h^{-1}$ $kg^{-1}$ | ref. |
|---|---|---|---|---|---|---|---|
| 0.7%Zn/ZSM-5(40)-IWI | impregnation | 11.7 | 5/95=$C_3H_8$/$N_2$, 550 ℃ | 0.5 | 0.42 | 2.35 | 17 |
| 0.17%Zn/ZSM-5(20)-IE(16 h) | ion exchange | 11.7 | 5/95=$C_3H_8$/He, 550 ℃ | 0.29 | 0.69 | 2.23 | 17 |
| 1.0%Zn/ZSM-5(20)-CVD/OX | chemical vapor deposition | 11.7 | 5/95=$C_3H_8$/He, 550 ℃ | 0.36 | 0.48 | 1.93 | 17 |
| 10%ZnO0.1%Pt/ZSM-5 | co-impregnation | 0.4 | 5/95=$C_3H_8$/$N_2$, 525 ℃ | 0.52 | 0.85 | 0.17 | 18 |
| 1.3%Zn/HZSM-5 | impregnation | - | 26.2/73.8= $C_3H_8$/He, 500 ℃ | 0.08 | 0.35 | - | 21 |
| 3c-ZnO-Y | atomic layer deposition | ~1.2 | 5/95=$C_3H_8$/Ar, 550 ℃ | 0.17 | 0.87 | 0.17 | 22 |
| 3c-ZnO-ZSM-5 | atomic layer deposition | ~1.2 | 5/95=$C_3H_8$/Ar, 550 ℃ | 0.83 | 0.1 | 0.10 | 22 |
| 4ZnO/TiZrO$_x$ | impregnation | 4.71 | 40/5/55=$C_3H_8$/$H_2$/$N_2$, 550 ℃ | 0.29 | 0.95 | 1.28 | 24 |
| 10%ZnO/deAl Beta | impregnation | 0.4 | 5/95=$C_3H_8$/$N_2$, 600 ℃ | 0.53 | 0.93 | 0.19 | 25 |
| 10%ZnO/ZSM-5 | impregnation | 0.6 | 5/95=$C_3H_8$/$N_2$, 600 ℃ | 0.92 | 0.64 | 0.33 | 26 |
| 15%ZnO0.1%Pt/γ-$Al_2O_3$ | impregnation | 3.3 | 28/28/44=$C_3H_8$/$H_2$/$N_2$, 600 ℃ | 0.35 | 0.94 | 1.04 | 27 |
| ZnO@NC/S-1(1.0) | *In-situ* carbonization | 0.9 | 14/14/72=$C_3H_8$/$H_2$/$N_2$, 600 ℃ | 0.55 | 0.87 | 0.40 | 31 |
| ZnO-S-1_3 | physical mixing | 7.9 | 40/60=$C_3H_8$/$N_2$, 550 ℃ | 0.31 | 0.87 | 2.04 | This work |
| ZnO-S-1_3 | physical mixing | 15.7 | 40/60=$C_3H_8$/$N_2$, 600 ℃ | 0.4 | 0.88 | 5.03 | This work |

PDH performance of the state-of-the-art Zn-containing catalysts and those developed in this study.