## [Peer Review File · Nature]

Manuscript Title: In-situ formation of ZnO_x species for efficient propane dehydrogenation

Editorial Notes:

Redactions – unpublished data

Reviewer Comments & Author Rebuttals

Reviewer Reports on the Initial Version:

===== Referees' comments =====

Referee #1 (Remarks to the Author):

This manuscript reports on a new synthesis of Zn⁺² ions supported on silicalite and other supports. The synthesis is suggested to be simple and the catalyst performance is concluded to exceed rates and yields better than commercial catalysts of Cr and Pt. The details of the experimental procedures, analysis and characterizations are incomplete. The conclusions are not convincing and need more and better quality data and analysis. This paper is not recommended at this time.

Comments:

1. The experimental procedures for synthesis and characterization are poorly described in the experimental section. The details of the synthesis and composition of the resulting catalyst are not given for example. The fitting of experimental methods and tables of the fit results are not given. Figures are shown, without detailed explanation or full comparison of all the results. Figure 1a, for example, shows some rates/g and some catalysts clearly have higher rates than others. However, no discussion is given as to why these different results occur. Are these due to differences in Zn loading, type of Zn species on the support or something else? The rates are better given as turnover rates (TOR), which would allow for better comparisons. Why is TiO₂ a better support than Al₂O₃, which is better than SiO₂? Why are the different silicalite samples better than SiO₂, and why do these differ from one another? These comparisons need to be quantitative and supported by data, rather than qualitative rationalization. Figure 1d shows periods or regeneration. The details of how this is done are not given. There is no loss of conversion, which indicates it doesn't need regeneration, so why do this? Since there is no loss of conversion, these results do not demonstrate regeneration once deactivated. The comparisons are made for Cr₂O₃/Na-Al₂O₃ and Pt. Commercially Cr is used at lower than 1 atm pressure, higher conversions (at about 650C) and the space time yield is determine by the endothermic temperature drop. As a result, those catalysts are run for 10 min and then regenerated (up to 60,000 times) to produce heat for the reheat the catalyst. Pt is usually stable for several days before regeneration with high rates and selectivity. So what is the composition of Pt, monometallic or a PtSn alloy? What was the selectivity, stability? So how can these Zn on silicalite be fairly compared? The synthesis method, Zn loading, catalytic results and characterizations on Al₂O₃, TiO₂ and ZrO₂ are very brief, with little understanding of how these compare to the other catalysts. The entire paper lacks detailed analysis and presentation in the manuscript.

2. The structure of Zn on SiO₂ is suggested to be a dimer. The evidence is a weak peak of Zn-O-Zn in Figure 2a, sample S_1. However, this could easily be a mixture of Zn-O-Si single site with a small amount of ZnO NPs. MCM-41, has excess ZnO in the sample, is not active, thus, the assignment of ZnO dimers is not convincing. Why are active sites produced on silicalite, but not on MCM-41 or SiO₂? If the amount of Zn in the synthesis is higher (and lower) on the good silicalite, is the higher shell Zn-O-Zn peak identical, or does the intensity change? The EXAFS fits are reported to be 3 for the starting catalyst; however, Zn⁺² compounds are generally 4 coordinate. Where are the results of the EXAFS fitting? Changes in sigma-squared can change the fit of the coordination number. How does the fit of the first shell Zn-O coordination compare in Q-space? How is a CN of 3 explained? What is the evidence of a dimer?

3. The change in the Zn XANES with H₂ is suggested to be due to reduction of Zn⁺². However, there are no stable Zn⁺¹ compounds, for example. The changes in XANES are often due to changes in coordination geometry, and ligand types. As the sample is heated in H₂, there is a decrease in the first shell Zn-O coordination number, Figure S8a. This would give rise to changes in the XANES. What is the evidence of Zn⁺² reduction? Yes, the coordination of Zn is changing, but why does this mean there is a change in oxidation state? Is there any literature that supports the assignment to Zn⁺¹ in these samples? The EXAFS is not consistent with metallic Zn, so what does it mean when Zn is reduced?

4. The EPR spectra in Figure 2c is suggested to be due to an electron trapped in an O vacancy. However, the fresh sample, is only slightly smaller than the reduced sample. If the O vacancy due to reduction is present, then the change in the Zn-O coordination number should directly correlate with the increase in the EPR signal. Figure S8c shows much larger losses in the Zn-O coordination than are present in the EPR. The EPR at each of the temperature of the EXAFS should be obtained. The EXAFS Zn-O coordination numbers should be fit and a correlation between the EPR and EXAFS be given. The same comparison of the XANES with EPR could also be made if, in fact, Zn is reducing and producing a trapped electron. If this is true, what does it mean structurally to have a vacancy which stabilizes a free electron? Vacancies are voids. If the electron is interacting and stabilized by the support, wouldn't it be expected to modify the properties of that electron and have a different spectral feature. Small amounts of EPR signals are common on high surface area supports, and are not often assigned to a specific structural feature.

5. If the ZnO_x is not a dimer, 3 coordinate and is reduced under reaction, then the modeling is not reliable and do not contribute to the understanding of these catalysts. Does the modeling suggest what is meant by reduced Zn⁺²? If one models other single size Zn⁺² catalysts are the energetics more favorable for this structure? Do the activation barriers agree with experiments? The kinetics of these catalysts have not been determined, which might validate the modeling.

6. It is not clear to me why the different supports are different in rate? How do they compare in selectivity, deactivation and structure? There is insufficient data to understand the performance and structure of these catalysts and compared to these to one another. If the authors understand the active site structure and can measure these, the the TOR for each catalysts can be determined. These should be the same and differ in the number of sites. If we can't compare the catalysts in this study, how can be compare the results for the standard catalysts? Where did the latter come from and how do these results compare to the literature?

The data in this manuscript does not support the conclusions. More quantitative and careful analysis is required. Thus, this manuscript is not recommended.

Referee #2 (Remarks to the Author):

A. Summary of the key results

The presented manuscript describes the research conducted by the authors on the development and structure-reactivity relationship of a ZnO based catalysts for dehydrogenation of light alkanes (ethane, propane and butanes) to alkenes as alternative to the commercially employed catalytic processes of direct alkane dehydrogenation Oleflex® and Catofin®. Several powerful structural characterizations techniques were used in the described work such as DRIFTS, EXAFS and EPR. A relevant study was done using deuterated feed components to observe the changes in the kinetics of the alkane dehydrogenation reaction and to assist in understanding of the mechanism of the reaction.

The authors synthesized their prospective catalyst materials using mesoporous supports to optimize dispersion and follow their structure-performance relationship to conclude that the concentration of surface bound hydroxyl groups is a major contributor to the catalyst performance.

The authors also have performed extensive computational effort to support their understanding of the reaction mechanism and attempt to fill out the knowledge gaps with mechanism DFT calculations for the possible reaction pathways.

B. Originality and significance: if not novel, please include reference

The work in the field of using Zn-based catalyst is not completely new as there are works preceding the author's work. This work improves the understanding of the catalyst system by using in situ characterization of the prepared catalysts. Authors mention the existing body of literature in passing in their introduction but do not discuss why and how their work complements the previously reported work.

One work that is conspicuously absent from the mentioned references is the work by Wan and Chu (J. Chem. Soc. Faraday Soc. 88(19), 2943, 1992) which is relevant to the described work and discusses kinetic findings in propane dehydrogenation over substantially similar catalyst to some of the examples in the manuscript.

C. Data & methodology: validity of approach, quality of data, quality of presentation

Approach. The authors use powerful methods for characterization of the catalysts, however their approach could be somewhat better designed in this reviewer's opinion. Their comparison to the industry methods for direct dehydrogenation of lower alkanes (Catofin® and Oleflex® processes) is rather simplified discussing their conversion performance comparison with a Catofin like catalyst (K-Cr₂O₃/Al₂O₃) alone. It would be prudent for the authors to discuss that outright conversion performance in the propane dehydrogenation system is somewhat secondary objective to suppressing side reactions such as cracking and coke formation. It is acknowledged in literature that per pass conversion can be higher at the expense of more side reactions and coking and more frequent catalyst regeneration (Asinger, F., *Monoolefins: Chemistry and technology*, 1968). Catalyst stability over multiple regeneration cycles and a compromise in cracking/per pass yield is how the industrial catalyst for this process are chosen.

It is highly preferable if the authors quote exactly their origin of 'industry relevant catalyst' as there are several commercial manufacturers of the Cr-based catalyst. It is possible that the catalyst they have obtained is optimized for particular industrial operating conditions.

The authors should have addressed directly the coke formation in the process as well as cracking byproduct formation as these are the two major issues faced in the industry, not catalyst outright conversion performance. In the manuscript supporting information, coke is determined by subtraction of the products and the feed they measure in their reactors. Considering the effort in completing the work, I can see significant value in the authors performing direct, rigorous experimental analysis of coke formation on their catalyst by using TGA, CO₂-measurements, etc. rather than reporting it as a calculation based analysis in their work. The same is true for the byproducts of propane dehydrogenation such as ethane, ethylene and methane formation. This side reaction cracking performance is a very important part of direct dehydrogenation frequently requiring additional separation in industrial processes. The authors do not comment on the cracking products whatsoever, while they are very important at drawing a correct understanding of a commercial catalyst candidate. The same applies to the formation of traces of MAPD (propadiene/methylacetylene) which are unacceptable for the use of the obtained propylene for polymerizations.

Another weakness in the approach is the comparison with the commercial catalyst at potentially inappropriate conditions. The Oleflex® commercial process using the Cr-catalyst uses very frequent regeneration due to fast coking of the catalyst, as frequently as every 15 minutes. The authors performed data collection after 3h on stream, which likely is well past the optimal window for commercial catalyst performance.

In reactions where the product distribution changes appreciably over time as well as conversion is challenging to characterize and should either include frequent, online GC data collection (e.g. every 2 minutes) or a cumulative sample collection over a preset time period that shows the integrated overall performance. I believe the authors should discuss their sample collection analysis in much more detail than provided in the manuscript and the supporting information.

I believe that the quality of data collected here is generally good specifically to their EXAFS and DRIFTS data. I found the lack of absorption isotherm measurements for their prepared catalysts disappointing; it would confirm they indeed succeeded in preparing the catalyst materials as anticipated. As discussed in the approach above, the description of the GC-data is not reported well enough; a thorough measurements with a table showing high carbon balance (at least 95% +) would be desirable.

The lack of any analysis of coke quantity, kinetics of formation and any characterization is sorely missing as this is one of the most significant challenges in the industrial use of direct dehydrogenation.

The computational analysis appears extensive and thorough, no remarks there.

Quality of presentation. In this reviewer's opinion, the quality of the presentation could be improved appreciably by significantly better comparison with the state of the art in industry, better comparison with the literature body on the subject and more clearly defined conclusions. The manuscript lacks cohesiveness in mating the experimental results from DRIFTS, EXAFS and the computational study. It feels that the manuscript sections are simply attached next to each other without significant integration that would benefit the potential reader.

D. Appropriate use of statistics and treatment of uncertainties

The use of statistics and uncertainties appears adequate.

E. Conclusions: robustness, validity, reliability

The conclusions in this manuscript are a mixed bag of effort and data interpretation varying from really good and thorough to inadequate. I found some of the studies quite thorough and well explained to lacking (GC and experimental analysis, coke analysis, comparison to industrial catalyst).

F. Suggested improvements: experiments, data for possible revision

Better comparison with industrial catalysis and the literature body on ZnO based dehydrogenation catalysts is needed. I would recommend experiment with excellent carbon balance, byproduct characterization, coke catalyst analysis and kinetics. At last this manuscript has to be better integrated between the describe different characterization techniques.

G. References: appropriate credit to previous work?

The credits appear adequate. I would recommend that authors take a look at the book mentioned earlier by Asinger and the Wan and Chu publication in J. Chem. Soc. Frm 1992 1992

H. Clarity and context: lucidity of abstract/summary, appropriateness of abstract, introduction and conclusions

I believe this manuscript could benefit from significant improvements in its structure. For example the title says 'exploiting simple redox chemistry', while after reading the manuscript it is not clear what is meant by that.

I believe the introduction could benefit from significant overhaul. The topic is not well introduced, the previous work is simply cited without any discussion of relevance and the industrial processes in current use are not mention to any appreciable degree. The introduction also does not fulfill its goal of setting the anticipated benefits of the described work.

The same possible improvements apply to the conclusions of the manuscript. The conclusions do not summarize the work well and put the emphasis on the DFT calculations alone.

==== End of comments =====

Author Rebuttals to Initial Comments:

Referee #1 (Remarks to the Author):

This manuscript reports on a new synthesis of Zn⁺² ions supported on silicalite and other supports. The synthesis is suggested to be simple and the catalyst performance is concluded to exceed rates and yields better than commercial catalysts of Cr and Pt. The details of the experimental procedures, analysis and characterizations are incomplete. The conclusions are not convincing and need more and better quality data and analysis. This paper is not recommended at this time.

Specific comment1

The experimental procedures for synthesis and characterization are poorly described in the experimental section. The details of the synthesis and composition of the resulting catalyst are not given for example. The fitting of experimental methods and tables of the fit results are not given.

Reply to comment 1

We apologize for omitting this important information in the main manuscript. We followed the advice of Nature: "The author should include "Methods" section at the end of the text" and have written a brief (probably too brief) experimental section, while further details of catalyst preparation and characterisation, catalytic tests, and DFT calculations have been provided in the sections "Synthesis of bare supports and catalysts", "Catalyst characterization", "Catalytic tests", and "DFT calculations" in the Supplementary Information, respectively. Our short description in the section "Methods" has perhaps provoked the criticism of the referee. To avoid any misunderstanding, we provide now all the experimental details in in the section "Methods". Some specific details related to DFT calculations are in the Supplementary Information.

Specific comment2

Figures are shown, without detailed explanation or full comparison of all the results. Figure 1a, for example, shows some rates/g and some catalysts clearly have higher rates than others. However, no discussion is given as to why these different results occur. Are these due to differences in Zn loading, type of Zn species on the support or something else?

Reply to specific comment 2:

We apologize if we were not clear enough when describing and discussing the results presented in Figures. This aspect has been considered upon revising the manuscript. We hope that our revisions have improved the manuscript in this regard.

Specific comment 3

The rates are better given as turnover rates (TOR), which would allow for better comparisons. Why is TiO₂ a better support than Al₂O₃, which is better than SiO₂? Why are the different silicalite samples better than SiO₂, and why do these differ from one another? These comparisons need to be quantitative and supported by data, rather than qualitative rationalization.

Reply to specific comment 3:

We agree with the referee that turnover frequency should be used for a proper comparison of different catalysts. Unfortunately, it is not possible when we test a physical mixture of ZnO and a support material due to the following reasons. We know that active sites are formed in the support *in situ* under PDH conditions, while ZnO is inactive. It is difficult to determine the concentration of catalytically active ZnO_x species formed because ZnO and the support with ZnO_x species cannot be separated from each other correctly. Therefore, it is difficult to calculate turnover rate in a correct way. When we used our dual-reactor approach for catalyst preparation (Extended data Figure 2a in the revised Supplementary Information), it was possible to avoid ZnO impurities in the catalysts. So-prepared S-1(H₂) and MCM-41(H₂) have comparable Zn loading (Extended Data Table 1 in the revised Supplementary Information), but differ in their activity by a factor of about 150 (Extended Data Figure 2c in the revised Supplementary Information). Thus, we think we can fairly compare those catalysts using the rate of propane formation.

As mentioned in the original manuscript, the kind of ZnO_x species (bi-atomic versus ZnO nanoparticles) determines catalyst activity. To answer why the kind of support is decisive for the formation of certainly structured ZnO_x species, we have carried out additional experiments. As seen in Figure R1, TiO₂, Al₂O₃, SiO₂ and Silicalite-1(S-1) showed totally different spectra in the OH stretching region. TiO₂ and S-1 showed two OH bands at about 3720 cm⁻¹ and lower wavenumber (about 3300 cm⁻¹ or about 3540 cm⁻¹), which could be attributed to isolated OH groups and hydrogen-bonded OH groups also called OH nests. The latter are important to capture Zn and *in situ* form active ZnO_x species, while SiO₂ only possesses isolated OH groups, which may be the anchor sites for ZnO nanoparticles. We have modified the manuscript accordingly.

Figure R1 The IR spectra of S-1_3, TiO₂, SiO₂ and Al₂O₃ in the OH stretching region and in situ DRIFT spectra of bare supports and physical mixture of ZnO -metal and a certain support material at 550 °C in 50% vol% H₂.

Specific comment 4

Figure 1d shows periods of regeneration. The details of how this is done are not given. There is no loss of conversion, which indicates it doesn't need regeneration, so why do this? Since there is no loss of conversion, these results do not demonstrate regeneration once deactivated.

Reply to comment 4:

The details have been given in the section "Catalytic test" in the original Supplementary Information. The reaction conditions have been provided in the caption of Figure 1d in original main text (Figure 3c in the revised main text).

Yes, you are right. Our catalyst did not show any visible deactivation within about 1 h on stream in these tests. The reason for catalyst "regeneration" was to check if catalyst on-stream stability and activity change after exposure to reducing and oxidising feeds. It is known that Pt-containing lose their performance after

reoxidation. Our Zn-S-1 catalyst prepared by physical mixing of ZnO and silicalite-1 showed high durability, which means the binuclear Zn species and the support are thermally stable, what is important from an industrial viewpoint.

Specific comment 5

The comparisons are made for Cr₂O₃/Na-Al₂O₃ and Pt. Commercially Cr is used at lower than 1 atm pressure, higher conversions (at about 650°C) and the space time yield is determined by the endothermic temperature drop. As a result, those catalysts are run for 10 min and then regenerated (up to 60,000 times) to produce heat for the reheat the catalyst. Pt is usually stable for several days before regeneration with high rates and selectivity. So what is the composition of Pt, monometallic or a PtSn alloy? What was the selectivity, stability? So how can these Zn on silicalite be fairly compared?

Reply to specific comment 5

We should mention that we did not synthesize Pt-based catalysts by ourselves. The data, we used in Figure 1b in the main text of the original manuscript, are from previously published studies. The corresponding references have also been provided. Both monometallic Pt- and PtSn-based catalysts are shown in this figure. The composition, selectivity, and conversion achieved over those catalysts are now provided in Extended Data Table 5 in the revised Supplementary Information. Importantly, a ZnO-S-1 physical mixture and an analogue of commercial K-CrO_x/Al₂O₃ were tested in same set-up under same conditions. Thus, their activity and selectivity can be directly compared (Figure 3 b in the revised manuscript).

Specific comment 6

The synthesis method, Zn loading, catalytic results and characterizations on Al₂O₃, TiO₂ and ZrO₂ are very brief, with little understanding of how these compare to the other catalysts. The entire paper lacks detailed analysis and presentation in the manuscript.

Reply to comment 6:

The catalysts were simply prepared by physical mixing of ZnO and a support material. The caption Figure 1a in the original manuscript had the following sentence “The rate of propene formation ($r(\text{C}_3\text{H}_6)$) at 550 °C over reduced ZnO-support mixtures.” The loading of ZnO has been provided.

To avoid any misunderstanding, we have modified the manuscript and provide new *in situ* DRIFTS results supporting our original conclusion about the importance of hydrogen-bonded OH groups for anchoring metallic Zn atoms (Extended Data Figure 31 in the revised Supplementary Information). This statement was further confirmed by analysing gas-phase products formed upon temperature-programmed heating of metallic Zn and different metal oxide supports. So-formed gas-phase products were analysed by an on-line mass spectrometer (Extended Data Figure 30 in the revised Supplementary Information).

In the revised manuscript, we prepared TiO₂(H₂), ZrO₂(H₂), TiZrO_x(H₂), and YZrO_x(H₂) according to our dual-bed reactor concept (Figure 2a). Zn loading in these samples is 1.5, 1.9, 2.0 and 1.2 wt%, respectively. The corresponding rates of propene formation are 0.63, 0.94, 1.38 and 1.12 mmol g⁻¹ min⁻¹, (Figure 3 d in the revised manuscript). This information is now available in the revised manuscript.

Specific comment 7

The structure of Zn on SiO₂ is suggested to be a dimer. The evidence is a weak peak of Zn-O-Zn in Figure 2a, sample S_1. However, this could easily be a mixture of Zn-O-Si single site with a small amount of ZnO NPs.

Reply to comment 7:

There are several reasons to think that we are not dealing with a mixture of Zn-O-Si and ZnO NPs. The first one is because the fit of the “weak” peak. One could make this “weak” peak stronger by weighting by k at higher power. However, if all features should be equally presented, it is better to use k^2 -weighting as most of researchers do in case of Zn) fits very well to one Zn neighbour, both in amplitude and phase (Extended Data Figure 9e). And the Debye-Waller factors (DW, σ^2) of both O and Zn shells are the same and in a typical range for covalently bonded molecular species at room temperature.

If we now start thinking about a mixture of Zn-O-Si, the spectrum of this species would not have a zero background after the O shell but the considerable background will be visible, see figure for the S_1 sample fit assuming Si in the second shell.

Figure R2 S_1 sample fit assuming Si in the second shell

The background from the Si is lower than the Zn contribution but is not zero, if we now add a Zn contribution over it we will not be able to achieve a fit in both amplitude and phase.

Second, if we look at the Zn-Zn bond distance in the binuclear Zn species in the S_1 spectrum, it is noticeably lower than the average Zn-Zn distance in ZnO (see footnote to the Extended Data Table 4). Third, due to splitting of the second shell in ZnO (see footnote to the Extended Data Table 4) we end up with high mean-square displacement of interatomic distances for this shell (Debye-Waller factors) in the fits containing ZnO, see e.g. the MCM-41 fit. This is not the case for the ZnO_x dimers where the DW factor of Zn is reasonably small and equal to that of the O shell.

Finally, 5.6ZnO/S-1 prepared by another method (see Methods in the main text) possesses isolated ZnO_x species but shows lower activity than S-1_1(H₂), i.e. 0.57 vs. 0.78 mmol g⁻¹ min⁻¹. Importantly, Zn loading in 5.6ZnO/S-1 is 5.6 wt%, while that in S-1_1(H₂) is only 1.2 wt%. Thus, the difference in terms of Zn-related turnover frequency (TOF) is even larger; 0.012 vs. 0.066 s⁻¹. A previous study (ACS Catal. 2014, 4, 1091–1098) with Zn/SiO₂ (amorphous silica) possessing isolated ZnO_x sites also reported low catalyst activity (TOF=0.0002 s⁻¹). Bulk (nanoparticles) ZnO in MCM-41(H₂) is not active, too. As our S-1_1(H₂) is highly active, it should have ZnO_x species, which are not isolated or ZnO particles. We have improved the manuscript in this regard.

Specific comment 8

MCM-41, has excess ZnO in the sample, is not active, thus, the assignment of ZnO dimers is not convincing.

Reply to specific comment 8

The referee's statement is not obvious for us. Firstly, we have to mention that the S-1(H₂) and MCM-41(H₂) possess comparable amount of Zn, which are 1.2 wt% and 0.98 wt%, respectively. The ICP results were summarized in the Extended Data Table 2 in the original version of Supplementary Information (Extended Data Table 1 in the revised Supplementary Information).

Secondly, according to our XAS analysis, MCM-41(H₂) contain large ZnO_x species. As this sample shows very low rate of propene formation (Extended Data Figure 2c), we can conclude that such species are not active. If S-1(H₂) also had such species, it would show low activity. Contrarily, it contains another species (binuclear Zn-oxo species) which is highly active. These results support our conclusion about the kind of active sites.

Specific comment 9

Why are active sites produced on silicalite, but not on MCM-41 or SiO₂?

Reply to specific comment 9:

To clarify this question, we have carried out addition DRIFT and temperature-programmed experiments. OH nests present in silicalite-1 seem to be able to catch only two Zn atoms to form bi-atomic ZnO_x species, while ZnO_x nanoparticles are formed upon reacting Zn with isolated OH groups. For details, please see our reply to your specific comment 6.

Specific comment 10

If the amount of Zn in the synthesis is higher (and lower) on the good silicalite, is the higher shell Zn-O-Zn peak identical, or does the intensity change?

Reply to specific comment 10:

As seen in the Extended Data Table 4, the coordination number in the second shell (and therefore the intensity, since the CN is directly proportional to the intensity) changes dramatically, although the amount of Zn which was introduced during the synthesis is same

Specific comment 11

The EXAFS fits are reported to be 3 for the starting catalyst; however, Zn⁺² compounds are generally 4 coordinate.

Reply to specific comment 11

The determined CN of 3 is another reason to believe in a different chemical nature of Zn species in S-1 compared to ZnO.

Specific comment 12

Where are the results of the EXAFS fitting?

Reply to specific comment 12:

Please see Extended Data Table 4 (The original version also contained this table to which we referred).

Specific comment 13

Changes in sigma-squared can change the fit of the coordination number.

Reply to specific comment 13

Yes, therefore it is really worth to look in the Extended Data Table 4 and observe there for all ZnO-containing samples high and roughly the same ($12-15 \times 10^{-3} \text{ \AA}^2$) Debye-Waller factors (DW, σ^2) which correspond well to the fact that the second shell in crystalline ZnO is distorted and Zn-Zn distances are unequal (CN(Zn-Zn)=6, $r(\text{Zn-Zn})=3.213 \text{ \AA}$, CN(Zn-Zn)=6, $r(\text{Zn-Zn})=3.250 \text{ \AA}$, see footnote to the Extended Data Table 4). On the other hand, DW factors for both O and Zn shells in the active S-1 sample are equal at ca. $7 \times 10^{-3} \text{ \AA}^2$ which is a typical value for samples measured at room temperature and is typical for molecular species with covalent bonds (in contrast to crystals with distorted symmetry such as ZnO). Note also that the second shell fit (with its reduced DW compared to the ZnO species) in the S-1 sample is nearly perfect (Extended Data Figure 9e). This is one more reason to believe that we here are not speaking about small contribution of ZnO but about a molecular complex with covalent bonds which is the proposed dimeric ZnO_x species.

Specific comment 14

How does the fit of the first shell Zn-O coordination compare in Q-space?

Reply to specific comment 14

Naturally, it compares well, since it is a product of the R-space fit (Extended Data Figure 10 in the revised Supplementary information). We have made Figure R3 for you, equivalent to Extended Data Figure 10, to check that. But since the Extended Data Figure 10e gives more information (amplitude and phase) and here you see only a product derived from these two we do not believe it reasonable to take the journal space to include that figure for the general reader (normally journals ask to avoid duplication of the data).

Figure R3: Comparison of the Fourier-filtered (r-range 1.0 and 3.2 Å) EXAFS spectra (q-space) with the corresponding fits: a MCM-41(H₂); b deAl-Beta(H₂); c deAl-MOR(H₂); d S-2(H₂); e S-1_1(H₂); f S-1_1(H₂)_H₂_100 °C; g S-1_1(H₂)_H₂_200 °C; h S-1_1(H₂)_H₂_300 °C; i S-1_1(H₂)_H₂_400 °C; j S-1_1(H₂)_H₂_500 °C. Note that the fitting region for the *in situ* data was smaller due to worse data quality which results in less perfect fits above 9 Å⁻¹.

Specific comment 15

How is a CN of 3 explained?

Reply to specific comment 15

CN of 3 means that there are 3 oxygen atoms at a distance of ca. 2.0 Å from a Zn atom. It corresponds well to a stable active site structure suggested and analysed by DFT.

Specific comment 16

What is the evidence of a dimer?

Reply to specific comment 16

Please, see our replies to your specific comments 7, 8, 11, 13,14, especially about bond distances and DW factors.

Specific comment 17

The change in the Zn XANES with H₂ is suggested to be due to reduction of Zn²⁺. However, there are no stable Zn⁺¹ compounds, for example.

Reply to specific comment 17

This is not entirely true. Although Zn¹⁺ compounds are air- and moisture-sensitive, they have been synthesized and isolated [10.1126/science.1101356; 10.1021/ja053819r; 10.1021/om300649q], and even used as homogeneous catalysts [10.1039/C1CC12461G]. Interestingly, all of them are binuclear Zn complexes and share a common structural motif which is a Zn-Zn bond (evident analogy with our binuclear active sites). Taking this analogy into account one may think that having a binuclear active site is not a coincidence, it is beneficial to have two Zn atoms close to each other. There is also an obvious analogy with known and proven redox-active binuclear Cu active sites of similar structure in zeolite matrices [10.1126/science.aan5630].

Nevertheless, we do not conclude that Zn²⁺ is reduced. There are no doubts that upon catalyst treatment in H₂, the former scavenges an oxygen atom from the coordination shell of Zn resulting in lowering CN which, in turn, changes both EXAFS and XANES spectra. During this process, each hydrogen atom yields an electron. For non-reducible metal oxides, the electrons are located in oxygen vacancies (Gianfranco Pacchioni, Numerical Simulations of Defective Structures: The Nature of Oxygen Vacancy in Non-reducible (MgO, SiO₂, ZrO₂) and Reducible (TiO₂, NiO, WO₃) Oxides, in J. Jupille and G. Thornton (eds.), Defects at Oxide Surfaces, Springer Series in Surface Sciences 58, DOI 10.1007/978-3-319-14367-5_1). The electron density can be partially distributed between neighbouring Zn²⁺ cations resulting in their formal "reduction".

Specific comment 18

The changes in XANES are often due to changes in coordination geometry, and ligand types.

Reply to specific comment 18

This theoretical possibility is true. However, we should note that the changes in the E₀ position (while the core atom keeps the same oxidation state and ligands are unchanged) are due to changed symmetry, i.e. forbidden transitions now become allowed or vice versa. Since the number of allowed transitions changes, we should observe new peaks or peaks disappear. None of phenomena are observed in our case, just shifts of existing peaks. Importantly, the ligands in our case stay the same. Thus, the changes in the nature of ligands as the reason for the peak shift can be excluded.

Specific comment 19

As the sample is heated in H₂, there is a decrease in the first shell Zn-O coordination number, Figure S8a. This would give rise to changes in the XANES. What is the evidence of Zn²⁺ reduction?

Reply to specific comment 19

We did not claim about Zn²⁺ reduction. Please, see our reply to your specific comment 17.

Specific comment 20

Yes, the coordination of Zn is changing, but why does this mean there is a change in oxidation state?

Reply to comment 20

We were probably not clear enough with our discussion. The shift of the position of the Zn K absorption edge to lower energy upon sample heating does not mean that Zn²⁺ must become Zn⁺¹. In general, if lattice oxygen is removed from a non-reducible metal oxide, electrons are typically trapped in so-formed anion vacancy (Gianfranco Pacchioni, Numerical Simulations of Defective Structures: The Nature of Oxygen Vacancy in Non-reducible (MgO, SiO₂, ZrO₂) and Reducible (TiO₂, NiO, WO₃) Oxides, in J. Jupille and G. Thornton (eds.), Defects at Oxide Surfaces, Springer Series in Surface Sciences 58, DOI 10.1007/978-3-319-14367-5_1). Some electron density can be transferred to neighbouring metal cations. This will result in an apparent reduction and can cause the shift observed by us for the ZnO/SiO₂ system. We have modified the manuscript to avoid any misunderstanding.

Specific comment 21

Is there any literature that supports the assignment to Zn⁺¹ in these samples?

Reply to comment 21

Again, we do not claim the formation of Zn¹⁺. Please see, our reply to your specific comments 19-20. A widest collection of Zn reference spectra known so far is published in [10.1107/S160057751900540X]. The collection contains 40 XANES spectra of Zn²⁺ containing reference materials coordinated to different atoms including O, S, N. In the whole collection there is only one record with E₀ less than that of the ZnO, and that is a Zinc protoporphyrin sample in which Zn is coordinated with nitrogen atoms. Therefore, we can

tell that all XANES studies performed on Zn reference compounds so far have demonstrated lower E_0 only in the case of less electronegative nearest neighbours than oxygen (i.e. higher electronic density of Zn). Thus, we believe that higher electron density at Zn^{2+} cations is also a reasonable explanation of E_0 shifts to lower energies observed in our case. If a contrary example from a Zn chemistry is known to the reviewer, we would be happy to learn it.

Specific comment 22

The EXAFS is not consistent with metallic Zn, so what does it mean when Zn is reduced?

Reply to specific comment 22

Please see our reply to your specific comments 17 and 21.

Specific comment 23

The EPR spectra in Figure 2c is suggested to be due to an electron trapped in an O vacancy. However, the fresh sample, is only slightly smaller than the reduced sample. If the O vacancy due to reduction is present, then the change in the Zn-O coordination number should directly correlate with the increase in the EPR signal. Figure S8c shows much larger losses in the Zn-O coordination than are present in the EPR. The EPR at each of the temperature of the EXAFS should be obtained. The EXAFS Zn-O coordination numbers should be fit and a correlation between the EPR and EXAFS be given. The same comparison of the XANES with EPR could also be made if, in fact, Zn is reducing and producing a trapped electron. If this is true, what does it mean structurally to have a vacancy which stabilizes a free electron? Vacancies are voids. If the electron is interacting and stabilized by the support, wouldn't it be expected to modify the properties of that electron and have a different spectral feature. Small amounts of EPR signals are common on high surface area supports, and are not often assigned to a specific structural feature.

Reply to specific comment 23

The presence of an electron trapped in an O vacancy in the fresh sample can be explained as follows. This sample was prepared upon treating ZnO and Silicalite-1 in an H_2 -containing (50 vol% in N_2) flow at $550^\circ C$. As a consequence, anion vacancies could already be generated. Some of them have been refilled after exposure to air at room temperature.

It should be stated that upon removal of oxygen from ZnO lattice, oxygen vacancies with different charge states can be formed with spin states $S=1/2$ and 1 depending where the two unpaired electrons are located (localized/delocalized). Only signal attributed to single-electron trapped oxygen vacancies with $S=1/2$ is EPR-active. For this reason, the Zn-O coordination number evident from EXAFS investigations is not directly correlated with the intensity of the EPR signal due to the formation of EPR silent species.

With regards to the comments that "small amounts of EPR signals are common on high surface area supports and are not often assigned to a specific structural feature"; we did not observe any EPR signal for the bare support (Extended Data Figure 11). Thus, the detected EPR signal in the Zn-containing silicalite-1 is related to ZnO.

Specific comment 24

If the ZnO_x is not a dimer, 3 coordinate and is reduced under reaction, then the modeling is not reliable and do not contribute to the understanding of these catalysts. Does the modeling suggest what is meant by reduced Zn^{+2} ? If one models other single size Zn^{+2} catalysts are the energetics more favorable for this structure? Do the activation barriers agree with experiments? The kinetics of these catalysts have not been determined, which might validate the modeling.

Reply to specific comment 24

We hope that our replies to your specific comments 7, 8, 11, 13,14 convinced you that the ZnO_x is a dimer.

Specific comment 25

It is not clear to me why the different supports are different in rate?

Reply to specific comment 25

The types of OH groups play an important role for anchoring and stabilizing different ZnO_x species. The mesoporous SiO_2 materials (such as MCM-41, SiO_2 , SBA-15) only possess isolated OH groups, which favors the formation of ZnO nanoparticles. Silicalite-1 possesses both isolated OH groups and OH nests but only latter species can react with Zn resulting in active binuclear species (EXAFS results in Figure 1e in revised manuscript). Such process was also confirmed by in situ DRIFTS (Figure 1c in the revised manuscript and Extended data Figure 31). Thus, Zn-containing catalysts with different supports could show significantly different rates of propene formation due to the different kinds of ZnO_x species.

Some relevant discussion is presented in the sections “OH groups govern the formation of ZnO_x” and “Nature of catalytically active ZnO_x species”.

Specific comment 26

How do they compare in selectivity, deactivation and structure?

Reply to specific comment 26

We did not compare the selectivity and deactivation behaviour of ZnO-MCM-41, ZnO-S-2 and ZnO-S-1 catalysts at high degree of propane conversion, because the latter catalyst showed about 25 or 5 times higher rate of propane formation than that of ZnO-MCM-41 or ZnO-S-2 catalysts.

We compared our catalysts with a commercial-like K-CrO_x/Al₂O₃ catalyst at a similar degree of propane conversion of 23% at the same temperature. As shown in Figure 3b and Figure R5 (Extended Data Figure 28 in the revised Supplementary Information), our catalyst showed slightly lower selectivity to propene and slightly higher amount of coke but 2.5 times higher space time yield of propene (STY) in comparison with the K-CrO_x/Al₂O₃ catalyst. The amount of coke in the spent K-CrO_x/Al₂O₃ and ZnO-S-1_1 is 0.04 wt% and 0.12 wt%, respectively. The absolute amount of coke in the spent K-CrO_x/Al₂O₃ and ZnO-S-1_1 is 0.12 mg and 0.144 mg, respectively.

Figure R5 a CO₂ evolution upon temperature-programmed oxidation of K-CrO_x/Al₂O₃ and ZnO-S-1_1 sample after 3 h on propane stream at 550 °C; b the ratio of the amount of formed coke to that of converted C₃H₈ during 3 h on propane stream. Reaction conditions: 550 °C, C₃H₈:H₂:N₂=2:1:2, 10 ml/min, 0.3 g K-CrO_x/Al₂O₃, 0.12 g ZnO-S-1_1.

Specific comment 27

There is insufficient data to understand the performance and structure of these catalysts and compared to these to one another. If the authors understand the active site structure and can measure these, the the TOR for each catalysts can be determined. These should be the same and differ in the number of sites. If we can't compare the catalysts in this study, how can be compare the results for the standard catalysts?

Reply to specific comment 27

Please, see our reply to specific comment 3.

Specific comment 28

Where did the latter come from and how do these results compare to the literature?

Reply to comment 28

The catalytic data of Pt-based catalysts in the original Figure 1b are from the literature (Extended Data Table 5 in the revised Supplementary Information). We have synthesized the K-CrO_x/Al₂O₃ catalyst according to the patent [V. Fridman, US Patent 8101541 B2, 2012, Sud-Chemie Inc]. The catalytic data of K-CrO_x/Al₂O₃ were obtained in the same setup we used for testing all Zn-containing samples.

Specific comment 29 and the final statement

The data in this manuscript does not support the conclusions. More quantitative and careful analysis is required. Thus, this manuscript is not recommended.

Reply to specific comment 29 and the final statement

We hope that our thorough reply to all the above specific comments and the revisions, we have made, will change your opinion.

Referee #2 (Remarks to the Author):

A. Summary of the key results

The presented manuscript describes the research conducted by the authors on the development and structure-reactivity relationship of a ZnO based catalysts for dehydrogenation of light alkanes (ethane, propane and butanes) to alkenes as alternative to the commercially employed catalytic processes of direct alkane dehydrogenation Oleflex® and Catofin®. Several powerful structural characterizations techniques were used in the described work such as DRIFTS, EXAFS and EPR. A relevant study was done using deuterated feed components to observe the changes in the kinetics of the alkane dehydrogenation reaction and to assist in understanding of the mechanism of the reaction.

The authors synthesized their prospective catalyst materials using mesoporous supports to optimize dispersion and follow their structure-performance relationship to conclude that the concentration of surface bound hydroxyl groups is a major contributor to the catalyst performance.

The authors also have performed extensive computational effort to support their understanding of the reaction mechanism and attempt to fill out the knowledge gaps with mechanism DFT calculations for the possible reaction pathways.

Reply to A. Summary of the key results

We thank the referee for acknowledging our experimental and theoretical efforts.

B. Originality and significance: if not novel, please include reference

The work in the field of using Zn-based catalyst is not completely new as there are works preceding the author's work. This work improves the understanding of the catalyst system by using in situ characterization of the prepared catalysts. Authors mention the existing body of literature in passing in their introduction but do not discuss why and how their work complements the previously reported work.

One work that is conspicuously absent from the mentioned references is the work by Wan and Chu (J. Chem. Soc. Faraday Soc. 88(19), 2943, 1992) which is relevant to the described work and discusses kinetic findings in propane dehydrogenation over substantially similar catalyst to some of the examples in the manuscript.

Reply to B. Originality and significance:

Sure, there are some previous studies focused on Zn-based catalysts as we have already mentioned in the Introduction part. Those catalysts usually used amorphous SiO₂, Al₂O₃, dealuminated Beta, Y zeolite or non-dealuminated HZSM-5 zeolite as the supports. They showed either low propane conversion or low selectivity to propene. For ZnO_x/SiO₂ catalyst with isolated ZnO_x species, the TOF value as low as about 0.8 h⁻¹ in [dx.doi.org/10.1021/cs401116p]. For the catalysts, which used zeolite as the supports, they show low selectivity to propene since Zn species and Brønsted acid sites can synergistically catalyze conversion of light alkanes to aromatics instead of olefins. [doi.org/10.1021/cs200441e and J. Catal., 192-202 (1998)]. Details about those catalysts are summarized in Extended Data Table 5 in the revised Supplementary Information. In comparison with the state-of-the-art catalysts in Extended Data Table 5, our binuclear Zn-oxo based catalyst outperformed all Zn-based catalysts in terms of space time yield and showed both high conversion of propane and selectivity. Moreover, none of the previously reported studies used our simple preparation method and elucidated the kind and the mechanism of formation catalytically active sites from ZnO and a support material. Not only zeolites, but also metal oxides with defect OH groups can be used for preparation of highly active catalysts.

As the referee requested, we have added more discussion in the main text in the introduction part.

Concerning the work of Wan and Chu (J. Chem. Soc. Faraday Soc. 88(19), 2943, 1992), we have read this paper carefully. The catalyst in this paper shows much lower rate of propene formation at the same temperature and the same feed composition (about 0.18 mmol g⁻¹ min⁻¹ vs 1.97 mmol g⁻¹ min⁻¹). It was also prepared through a simple impregnation method. As also proven in our study, such preparation method results in a less active catalyst.

In addition, since the hydrogen-bonded OH groups (OH nests) play an important role in capturing Zn and forming binuclear Zn-oxo species rather than the supports, it is important to develop the support with such OH defects. As shown in our paper in the Extended Data Figure 6a, when synthesis conditions of silicalite-1 were changed, the amount of OH nests could be differed. The OH nests-free sample also could be synthesized in the presence of Na⁺, which was confirmed by DRIFTS in Extended Data Figure 6c in the revised Supplementary Information. However, when we physically mixed such support and ZnO, it only showed low rate of propene formation of about 0.01 mmol g⁻¹ min⁻¹. Thus, we concluded that the OH nests act as anchoring sites for the active Zn species.

C. Data & methodology: validity of approach, quality of data, quality of presentation approach.

The authors use powerful methods for characterization of the catalysts, however their approach could be somewhat better designed in this reviewer's opinion. Their comparison to the industry methods for direct dehydrogenation of lower alkanes (Catofin® and Oleflex® processes) is rather simplified discussing their conversion performance comparison with a Catofin like catalyst (K-Cr₂O₃/Al₂O₃) alone. It would be prudent for the authors to discuss that outright conversion performance in the propane dehydrogenation system is somewhat secondary objective to suppressing side reactions such as cracking and coke formation. It is acknowledged in literature that per pass conversion can be higher at the expense of more side reactions and coking and more frequent catalyst regeneration (Asinger, F., *Monoolefins: Chemistry and technology*, 1968). Catalyst stability over multiple regeneration cycles and a compromise in cracking/per pass yield is how the industrial catalyst for this process are chosen. It is highly preferable if the authors quote exactly their origin of 'industry relevant catalyst' as there are several commercial manufacturers of the Cr-based catalyst. It is possible that the catalyst they have obtained is optimized for particular industrial operating conditions.

Reply to C. Data & methodology: validity of approach, quality of data, quality of presentation approach. A commercial-like K-CrO_x/Al₂O₃ catalyst was synthesized according to the patent [V. Fridman, US Patent 8101541 B2, 2012, Sud-Chemie Inc]. The catalytic data of K-CrO_x/Al₂O₃ were obtained in the same setup we used for catalytic tests with other Zn-containing catalysts. Extended Tables 6 and 7 contain propane conversion, the selectivity to propene, cracking products and coke as well as carbon balance every 4 minutes in a 3 hours test.

Specific comment C1

The authors should have addressed directly the coke formation in the process as well as cracking byproduct formation as these are the two major issues faced in the industry, not catalyst outright conversion performance. In the manuscript supporting information, coke is determined by subtraction of the products and the feed they measure in their reactors. Considering the effort in completing the work, I can see significant value in the authors performing direct, rigorous experimental analysis of coke formation on their catalyst by using TGA, CO₂-measurements, etc. rather than reporting it as a calculation based analysis in their work. The same is true for the byproducts of propane dehydrogenation such as ethane, ethylene and methane formation. This side reaction cracking performance is a very important part of direct dehydrogenation frequently requiring additional separation in industrial processes. The authors do not comment on the cracking products whatsoever, while they are very important at drawing a correct understanding of a commercial catalyst candidate. The same applies to the formation of traces of MAPD (propadiene/methylacetylene) which are unacceptable for the use of the obtained propylene for polymerizations.

Reply to specific comment C1

Coke formation is very important for the PDH process and was considered in our original study. We have provided the selectivity to coke.

As the referee requested, we prepared Extended Data Tables 6 and 7 containing detailed information about propane conversion, product selectivity including coke, and carbon balance during 3 hours on stream (every 4 minutes) in PDH over ZnO-S-1 and K-CrO_x/Al₂O₃ respectively. For the ZnO-S-1 catalyst, the selectivity to cracking products (methane, ethene and ethane) is below 2%, while the selectivity to coke is below 8%. The carbon balance is above 98%. The commercial-like K-CrO_x/Al₂O₃ showed slightly lower cracking selectivity and coke selectivity.

As the referee asked, we have carried out temperature-programmed oxidation tests with spent catalysts to analyse the coke amount (Figure R5 and Extended Data Figure 28 in the revised Supplementary Information). Our catalyst showed slightly higher ratio of coke amount to the amount of converted C₃H₈ during 3 h on propane stream but 2.5 times higher space time yield of propene.

Figure R5 a CO₂ evolution upon temperature-programmed oxidation of K-CrO_x/Al₂O₃ and ZnO-S-1_1 sample after 3 h on propane stream at 550 °C; b the ratio of the amount of formed coke to that of converted C₃H₈ during 3 h on propane stream.

Specific comment C2

Another weakness in the approach is the comparison with the commercial catalyst at potentially inappropriate conditions. The Oleflex® commercial process using the Cr-catalyst uses very frequent regeneration due to fast coking of the catalyst, as frequently as every 15 minutes. The authors performed data collection after 3h on stream, which likely is well past the optimal window for commercial catalyst performance.

Reply to specific comment C2

We agree with you that our reaction conditions are not same as in the above-mentioned commercial process. Nevertheless, we have compared our catalyst with an industrial analogue of K-CrO_x/Al₂O₃ in same set-up under same conditions. As seen in Extended Tables 6 and 7, our catalyst is less selective, particularly during first 20 minutes on stream. The difference becomes less pronounced with rising time on stream. Importantly, our catalyst shows about 2.5 times higher productivity. We would also like to mention that we did not provide an industry-ready catalyst.

Specific comment C3

In reactions where the product distribution changes appreciably over time as well as conversion is challenging to characterize and should either include frequent, online GC data collection (e.g. every 2 minutes) or a cumulative sample collection over a preset time period that shows the integrated overall performance. I believe the authors should discuss their sample collection analysis in much more detail than provided in the manuscript and the supporting information.

Reply to specific comment C3

We are sorry for omitting this important information. We used an on-line GC for analysing outlet gases every 4 minutes. Maybe we did not clearly describe this procedure in the original section "Catalytic test" in the Supplementary Information. We have now provided this information in the emended Supplementary Information.

Specific comment 4

I believe that the quality of data collected here is generally good specifically to their EXAFS and DRIFTS data. I found the lack of absorption isotherm measurements for their prepared catalysts disappointing; it would confirm they indeed succeeded in preparing the catalyst materials as anticipated. As discussed in the approach above, the description of the GC-data is not reported well enough; a thorough measurements with a table showing high carbon balance (at least 95%+) would be desirable.

Reply to comment C4

We thank the referee for the positive evaluation of our data. The XRD and N₂ absorption-desorption characterization results are now shown in the Extended data Figures 4-5 and the surface area of supports is provided in Extended Data Table 2.

As the referee requested, the carbon balance during PDH over each catalyst we tested for long-time stability is now presented in the corresponding figures, for example, in Extended Data Figure 27 and 32, and Extended Data Tables 6-7. Except ZnO-deAl Beta (about 94%), the carbon balance of other samples is above 95%.

Specific comment C5

The lack of any analysis of coke quantity, kinetics of formation and any characterization is sorely missing as this is one of the most significant challenges in the industrial use of direct dehydrogenation.

Reply to specific comment C5

The amount of coke formed during 3 hours on propane stream was determined from temperature-programmed oxidation tests (Figure R5 and Extended Data Figure 28 in the revised Supplementary Information). The amount of coke in K-CrO_x/Al₂O₃ and ZnO-S-1_1 is 0.04 wt% and 0.12 wt%, respectively. The absolute amount of coke in the spent K-CrO_x/Al₂O₃ and ZnO-S-1_1 is 0.12 mg and 0.144 mg, respectively.

Specific comment C6

The computational analysis appears extensive and thorough, no remarks there.

Quality of presentation. In this reviewer's opinion, the quality of the presentation could be improved appreciably by significantly better comparison with the state of the art in industry, better comparison with the literature body on the subject and more clearly defined conclusions. The manuscript lacks cohesiveness in mating the experimental results from DRIFTS, EXAFS and the computational study. It feels that the manuscript sections are simply attached next to each other without significant integration that would benefit the potential reader.

Reply to specific comment C6

We thank the referee for the suggestions. We have modified the structure of the paper for clarity.

D. Appropriate use of statistics and treatment of uncertainties

The use of statistics and uncertainties appears adequate.

Reply to D. Appropriate use of statistics and treatment of uncertainties

We thank the referee for her/his positive statement.

E. Conclusions: robustness, validity, reliability

The conclusions in this manuscript are a mixed bag of effort and data interpretation varying from really good and thorough to inadequate. I found some of the studies quite thorough and well explained to lacking (GC and experimental analysis, coke analysis, comparison to industrial catalyst).

Reply to E. Conclusions: robustness, validity, reliability

We have modified the conclusions for clarity. Concerning to GC and coke analysis as well comparison with industrial catalysts, please see our reply to your specific comments C1 and C3.

F. Suggested improvements: experiments, data for possible revision

Better comparison with industrial catalysis and the literature body on ZnO based dehydrogenation catalysts is needed. I would recommend experiment with excellent carbon balance, byproduct characterization, coke catalyst analysis and kinetics. At last this manuscript has to be better integrated between the describe different characterization techniques.

Reply to F. Suggested improvements: experiments, data for possible revision.

As the referee requested, we provide now some relevant information. Please, see our reply to your specific comments C1 and C3.

G. References: appropriate credit to previous work?

The credits appear adequate. I would recommend that authors take a look at the book mentioned earlier by Asinger and the Wan and Chu publication in J. Chem. Soc. Frm 1992 1992

Reply G. References: appropriate credit to previous work?

For the recommended literatures, we have read the recommended literature carefully and cited the paper of Wan and Chu in the revised manuscript.

H. Clarity and context: lucidity of abstract/summary, appropriateness of abstract, introduction and conclusions

I believe this manuscript could benefit from significant improvements in its structure. For example the title says 'exploiting simple redox chemistry', while after reading the manuscript it is not clear what is meant by that.

Reply to H. Clarity and context: lucidity of abstract/summary, appropriateness of abstract, introduction and conclusions.

We have changed the title and the structure of the manuscript.

H. Clarity and context: lucidity of abstract/summary, appropriateness of abstract, introduction and conclusions

I believe the introduction could benefit from significant overhaul. The topic is not well introduced, the previous work is simply cited without any discussion of relevance and the industrial processes in current use are not mention to any appreciable degree. The introduction also does not fulfill its goal of setting the anticipated benefits of the described work.

The same possible improvements apply to the conclusions of the manuscript. The conclusions do not summarize the work well and put the emphasis on the DFT calculations alone.

Reply to H. Clarity and context: lucidity of abstract/summary, appropriateness of abstract, introduction and conclusions

As the referee suggested, we have modified the manuscript and hope that our revisions will change the referee's opinion.

Reviewer Reports on the First Revision:

===== Referees' comments =====

Referee #1 (Remarks to the Author):

This manuscript describes a preparation method for Zn⁺² oxides on silicalite, SiO₂ and other oxide supports, which are used for propane dehydrogenation. This is suggested to be potentially useful for commercial replacement of CrOx catalysts in the Catofin process. The primary advantage is that the new catalyst has 2-3x higher rate than the commercial catalyst. In addition, the authors characterize the active state of the catalyst. Several of the characterizations require more rigorous and critical analysis and several conclusions are known and others are not established by the results. Thus, this manuscript is not recommended.

Comments:

1. One of the primary conclusions is that Zn ion sites are stabilized by the hydroxyl groups by the support. This has previously been shown by Raman spectroscopy, see Ref. 25. The XANES and EXAFS in Ref 25 are very similar to those in Figure 1e and 1f of this manuscript despite the synthesis by different methods. Thus, the catalyst appears to be very similar to other Zn catalysts in the literature.
2. The active site is prepared by treatment of ZnO and the support at high T in H₂. This is suggested to be due to reduction to Zn metal and oxidation at the Si-OH sites. However, dispersion of oxides onto supports has been extensively studied for preparation of MoOx and WOx catalysts, and for the preparation of Co on SiO₂ propane dehydrogenation catalysts, see Ref 19. In Ref 19 CoO was heated and formed low coordinate Co⁺² on the support, which became active for PDH after treatment.
3. The formation of active catalytic centers from CoO precursors in Ref 19 was NOT thought to form by reduction to Co metal since this occurs at much higher T. In this manuscript there is no convincing evidence the ZnO is reduced to Zn which become volatile and is oxidized by the support to the active site. On page 4 it is stated that "we can conclude that metallic Zn atoms originated from the ZnO upon H₂ treatment react with the support..." None of the spectroscopy suggests that ZnO can be reduced to metallic Zn at these temperatures. On page 6, the conclusion of Zn metal is suggested from the low CN of 2 for Zn-O and Zn-Zn of 9.2. The fits of the Zn-O however, are low. In Extended Data in T4, the delta sigma squared (ss) is much lower than for the other Zn-O fits. Low ss values lead to low CN's. If the CN were 2, then about half the Zn in this sample is metallic. With a CN of 9.2, the true metallic CN would be 18, which is not possible. Also, the Zn-Zn distance is not consistent with metallic Zn, but more similar to ZnO. Many of the fits in ED T4 need to be fit better. The ss values for the high temperatures are also too low. There are procedures to determine the ss values and it is not just getting a good fit in R-space.
4. On page 6, it is suggested that the active site is binuclear. EXAFS is not a good method to determine if it is a monomolecular, bimolecular, trimolecular or other size. Single sites were previously proposed by poisoning stoichiometries for Zn⁺² (ref 25) and EPR for Co⁺² (ref 19). While it is possible that are exclusively binuclear sites as suggested, this conclusion is not supported by the data.
5. In ref 25, the Zn-O coordination number of the active catalyst decreased from 4 to 3 at high temperature in hydrogen as shown in this study. In 25, the true coordination of the latter thought to have 3 Zn-O and also contain a Zn-H, which would not be detected by EXAFS. In similar Ga PDH catalysts Ga-H has been detected by Raman and Infrared spectroscopy and the loss of Ga-O bonds correlates with Ga-H formation and not defect formation as suggested here.
6. The EPR in this study is suggested to results form a free electrons trapped at a reduced Zn vacancy site. The EPR signal is present in the as prepared sample, which should have not vacancies. The signal increase with the preparation treatment. However, the number of sites is not quantitatively compared to the intensity gain in the EPR. The EPR single has not been modeled to identify where the unpaired electron is located. Is it from the Zn ion, O ions, supports, adjacent to the two Zn if it is binuclear? It seems more likely that the signal is from the larger number of ZnO sites that are formed as the bulk oxide is dispersed on the support. It is not obvious that these are defect sites from the spectrum.
7. On page 8 it is suggested that OH group activate H₂. H-D exchange of support OH sites by D₂ on Pt

catalysts have long been thought to be due to H atom spillover from the metal. However, careful studies by the Bourdard group have shown that if one traps the D₂O, which is present in the D₂, at liquid N₂ temperatures, there is no support OH-OD exchange, and support OH groups do NOT activate H₂. The IR in this study does not indicate that they removed all the D₂O present in the D₂. Thus, the OD peaks are likely due to exchange with D₂O, rather than D₂.

8. Page 9 makes the case that these catalysts have potential for commercial applications. The Catofin process is a large capital-intensive unit, and the largest cost is separation of propylene from propane. The catalyst operates at the maximum possible temperature to obtain the highest conversion (above 50%) to minimize separation cost. This endothermic reaction leads to temperature drops and the conversion is quenched due to the equilibrium at the outlet temperature. Since the catalyst operates at equilibrium, higher rates do not lead to improved performance and higher yields. With the CrOx catalyst operation at higher T leads to lower selectivity. Process improvement to give higher space time yields and selectivity have been obtained by adding a heat generating material in the catalyst to increase the conversion while keeping the lower operating temperature. The CrOx catalysts has sufficient rate to give higher space time yields. Equilibrium limits the space time yields commercially. The limiting factors are selectivity and maximum operating temperature. The endothermic temperature drop controls the yields that any catalyst can get. In this paper, at 23% conversion a selectivity of 90-92 is lower than the CrOx catalyst. This alone would not lead to commercial application. The reaction temperatures are also at 550C, which is about 75C too low for commercial operations. At higher T one gets more thermal cracking, so this Zn catalyst would be expected to have even lower selectivity. If the catalyst can operate at 650C, 50+% conversion with a selectivity of greater than 95% (and be regenerated 30,000 times), then this might replace the CrOx catalyst, however, the increase would be approximately what the CrOx catalyst is and not represent a step change in technology. For PDH, the rate of the catalyst is NOT the limiting factor of performance.

The materials reported in this manuscript don't appear to be much different from those previously reported for Zn PDH catalysts. Many of the conclusions are known or not supported by the results, thus this is paper is not recommended.

Referee #2 (Remarks to the Author):

A. In my opinion, the authors have significantly improved the manuscript since its first iteration. The manuscript is now cohesive to read between different sections, some additional data was added and critical issues have been addressed.

B. The work is of high significance as it pertains to the highest volume chemicals manufactured worldwide. The work is reasonably original and the findings would be of interest to the scientific community in industrial chemistry, chemical engineering and materials characterization.

C. In this reviewer's opinion, the standout feature of the article is the thorough characterization methods for the catalyst and surface intermediates using excellent characterization instrumental and experimental tools. The validity of the approach is excellent, the data quality is sufficient and the presentation is easier to follow. The integrated DFT computational mechanics models and discussion contribute to the value of the manuscript.

D. The use of statistics and treatment of uncertainties appears acceptable.

E. It appears that the authors have put together an impressive multi-institutional effort to resolve a complicated scientific problem. In my opinion, the wealth of data was put to good use and the conclusions are well supported. This reviewer did not notice issues that would cause validity or reliability concerns.

F. No. The original manuscript had some serious issues that have been well addressed in this revision.

G. References on previous work on the subject are adequate.

H. The clarity and context are fine after the revision. This reviewer would prefer for the authors to

improve upon the abstract a bit. For example, comments about the superiority of the studied catalyst do not really belong there, as this is still a laboratory, experimental system.

Note: I believe on line 36 the authors likely meant Cr(VI) instead of Cr(IV).

==== End of comments =====

Author Rebuttals to First Revision:

Referee #1 (Remarks to the Author):

This manuscript describes a preparation method for Zn²⁺ oxides on silicalite, SiO₂ and other oxide supports, which are used for propane dehydrogenation. This is suggested to be potentially useful for commercial replacement of CrO_x catalysts in the Catofin process. The primary advantage is that the new catalyst has 2-3x higher rate than the commercial catalyst. In addition, the authors characterize the active state of the catalyst. Several of the characterizations require more rigorous and critical analysis and several conclusions are known and others are not established by the results. Thus, this manuscript is not recommended.

Reply to the general statement

The criticism of the reviewer was probably provoked by the unclear presentation and discussion of our results. The highlight of our study is NOT the catalyst. The primary advantages are (i) a simple method for preparation of highly active ZnO-containing catalyst based on ZnO and commercially available oxidic supports and, most importantly, (ii) the fundamentals enabling preparation of supported ZnO_x structures in a purposeful manner. The method is inexpensive, controllable, and easily scalable. In contrast with previous studies, we demonstrate that supported ZnO_x species can be formed from ZnO and an oxidic support, when these both components are simply treated in H₂ or CO above 500°C. The active sites are in situ generated in a reactor filled with a physical ZnO-support mixture or with a support with an on-top ZnO layer. This in situ method does not require expensive chemicals, wet chemistry, precise control of conditions. Filtering, washing, drying and calcination steps are no longer relevant. The method can be applied at the industrial site where it is to be used thereby massively decreasing associated costs and complexity. In certain cases, it may change the plant economy making otherwise unviable plants viable. This is the real main highlight of the work as is also reported in the synopsis.

The fact that the developed catalyst outperforms an analogue of commercial K-CrO_x/Al₂O₃ catalyst in terms of propene productivity further supports commercial attractiveness of our approach.

As written in our reply to the specific comments, we have applied the state-of-the-art methods for catalyst characterisation and the corresponding evaluation procedures to draw our conclusions. The main conclusions of our study are unknown. Please, read our point-by-point reply to your specific comments.

Comments

1. One of the primary conclusions is that Zn ion sites are stabilized by the hydroxyl groups by the support. This has previously been shown by Raman spectroscopy, see Ref. 25. The XANES and EXAFS in Ref 25 are very similar to those in Figure 1e and 1f of this manuscript despite the synthesis by different methods. Thus, the catalyst appears to be very similar to other Zn catalysts in the literature.

Reply to comment 1

The primary novelty of our study is related to the method of catalyst preparation and the fundamentals for creation of specific ZnO_x structures. Please, see our reply to your general statement.

Concerning the hydroxyl groups, we proved that the kind of hydroxyl groups is an important factor affecting the structure of ZnO_x species and accordingly their activity in the dehydrogenation of propane to propene. In general, there are very limited possibilities to bind Zn (any metal) cations/complexes in oxidic materials apart from via OH groups, therefore similarities are to be expected.

We are surprised by the statement "the catalyst appears to be very similar to other Zn catalysts in the literature" when considering that our ZnO-silicalite-1 catalyst shows about 300 times higher TOF (238 h⁻¹ vs. ~0.8 h⁻¹) value than the single-site Zn²⁺ in Ref. 25 mentioned by the referee (Ref. 26 in the revised manuscript). Thus, we do not think that the active ZnO_x species or their local geometry in our catalyst are same as those of ZnO_x in other Zn-containing catalysts described in the literature.

The XANES and EXAFS data in Ref. 25 (Ref. 26 in the revised manuscript) and in the present manuscript are indeed very similar. However, the authors of this previous study, for some reason, decided not to analyse and discuss the second shell, which they also have in their spectra at 3 Å uncorrected distance (therefore we arrived at different conclusions with respect to the nuclearity of the ZnO_x sites). As they did not provide fits and their reasons for not analysing the second shell, there is no real basis for further discussion. We have already shown that it is important to fit the second shell and Zn fits perfectly during the previous revision round, therefore we will not repeat it here.

2. The active site is prepared by treatment of ZnO and the support at high T in H₂. This is suggested to be due to reduction to Zn metal and oxidation at the Si-OH sites. However, dispersion of oxides onto supports has been extensively studied for preparation of MoO_x and WO_x catalysts, and for the preparation of Co on SiO₂ propane dehydrogenation catalysts, see Ref 19. In Ref 19 CoO was heated and formed low coordinate Co⁺² on the support, which became active for PDH after treatment.

Reply to comment 2

It is not obvious why the reviewer asks about dispersion of oxides onto support. We have, for the first time, introduced a simple method for preparation of supported ZnO_x species (please, see our reply to the general statement) and elucidated the fundamentals required for purposeful preparation of active catalysts. The method was additionally validated for synthesis of Ga-containing catalysts (Figure 3c in the revised main text).

To the best of our knowledge, all up to now reported MoO_x-, WO_x- or CoO_x- catalysts have been prepared through typical methods, like impregnation, atomic layer deposition, anchoring etc. Why are these papers detrimental for the novelty of our study?

The fact that non-reduced CoO_x-based catalysts have an induction period in the PDH reaction (*Journal of Catalysis* 381 (2020) 482–492; *Catalysis Communications* 60 (2015) 42–45; *Applied Surface Science* 441 (2018) 688–693; *Journal of Catalysis* 383 (2020) 77–87) simple means that oxidized CoO_x species are less active than their reduced counterparts. In contrast to our approach, CoO_x species have been initially deposited on supports through one of the methods mentioned above. Moreover, we would like to notice that a big difference in the melting points of Zn and Co (419.5 °C vs. 1495 °C) does not justify a direct comparison of these two metals with respect to the mechanism of active site formation.

3(i). The formation of active catalytic centers from CoO precursors in Ref 19 was NOT thought to form by reduction to Co metal since this occurs at much higher T. In this manuscript there is no convincing evidence the ZnO is reduced to Zn which become volatile and is oxidized by the support to the active site. On page 4 it is stated that “we can conclude that metallic Zn atoms originated from the ZnO upon H₂ treatment react with the support...” None of the spectroscopy suggests that ZnO can be reduced to metallic Zn at these temperatures.

Reply to comment 3(i)

Probably, we were not clear enough when describing our preparation method. Please, read the below experimental findings supporting our conclusion.

1. Although ZnO and the support are physically separated (see Extended Data Figure 2a), Zn was determined in the downstream located support.

2. Zn is not present in the downstream located support if ZnO and the support were treated (see Extended Data Figure 2a) under oxidizing or inert conditions (see Extended Data Table1).

3. Zn species appear in the downstream located support if ZnO and the support were treated (see Extended Data Figure 2a) under reducing conditions (see Extended Data Table1).

4. When a physical mixture of metallic Zn and the silicalite-1 support were treated in an inert gas, Zn was identified in the support layer and H₂ could be determined by an online mass spectrometer (see Extended Data Figure 3).

5. In situ DRIFT measurements proved the participation of OH groups in the formation of ZnO_x species (Figure 1c).

All these data had already been mentioned in the original manuscript.

3(ii) On page 6, the conclusion of Zn metal is suggested from the low CN of 2 for Zn-O and Zn-Zn of 9.2. The fits of the Zn-O however, are low. In Extended Data in T4, the delta sigma squared (ss) is much lower than for the other Zn-O fits. Low ss values lead to low CN's.

Reply to comment 3(ii)

First of all, we did not conclude the presence of metallic Zn species in the MCM-41(H₂) catalyst. Second, we agree about the fact that SS is highly correlated with CN. For the reason of better comparability, we fixed SS for the fit of MCM-41(H₂) to $7.3 \times 10^{-3} \text{ \AA}^2$ which is the most common value seen at room temperature. However, we must stress that by taking the same SS at the same temperature, we assume that the static disorder part of the SS is also the same. This assumption is not necessarily true as the structure is quite different. Therefore, the structural disorder may also be different that was the real reason why we allowed variation of the SS among different samples.

3(iii) If the CN were 2, then about half the Zn in this sample is metallic.

Reply to comment 3(iii)

We should notice here that we do not write "metallic". It may be highly defect oxide, e.g., similar to the case of e.g., CeO₂.

3(iv) With a CN of 9.2, the true metallic CN would be 18, which is not possible.

Reply to comment 3(iv)

We did not conclude the presence of metallic Zn species in the MCM-41(H₂) catalyst. We should draw the reviewer's attention to the fact that two independent SS were used for the fits of O and Zn shells. Therefore, after factoring in the corresponding SS value for each shell, the numbers make sense.

3(v) Also, the Zn-Zn distance is not consistent with metallic Zn, but more similar to ZnO.

Reply to comment 3(v)

We did not conclude the presence of metallic Zn species in the MCM-41(H₂) catalyst.

3(vi) Many of the fits in ED T4 need to be fit better.

Reply to comment (3vi)

Unfortunately, the reviewer did not explain what "better" means. Although, it is a very questionable approach to apply the same SS for fitting a priori unknown structures (which may have different static disorder component) we refitted the spectra of MCM-41(H₂), 5.6Zn/S-1, deAl-MOR(H₂) and S-2(H₂) with an SS value in the oxygen shell fixed to $7.3 \times 10^{-3} \text{ \AA}^2$ (the most commonly observed). The SS for both shells in the fits were kept the same in case of clusters of low nuclearity. For bulk-like nanoparticulate ZnO species, it is not possible to use the same SS for both shells since the static disorder in these structures is clearly different for Zn-O and Zn-Zn contacts in the ZnO crystal structure.

3(vii) The ss values for the high temperatures are also too low.

Reply to comment 3(vii)

We have to disagree here. Actually, the SS values are in the normal to high range. Unfortunately, the reviewer does not state, which SS values should be according to her/his opinion. Therefore, we can argue only by comparing with literature. According to Scott Calvin (XAFS for Everyone, Taylor & Francis Inc; Illustrated Edition (20. Mai 2013), ISBN-10 : 1439878633), "Typical values range from about 0.002 to 0.03 \AA^2 ". We are reporting 0.007 – 0.014 \AA^2 for the temperature series. If we now compare with the references exemplified by the reviewer, in Ref. 25 (Ref. 26 in the revised manuscript) SS of 0.001 is reported (and

0.006 for the measurement at 550 °C) – that would indeed be much too low (and in Ref. 19 (Ref. 20 in the revised manuscript) and those authors decided not to report SS at all).

Please see also SS plotted vs. Temperature for our heating series (Fig. R1 left). There are not many fundamental works known to us which review temperature dependence of the SS for non-monoatomic compounds, we could find such dependence for AgI (Fig. R1 right) [DOI: 10.1107/S0909049597006900], again our data fall nicely in the same region, even a little bit higher which is normal considering that we are not dealing with well-defined crystalline material.

Figure 8
MSRDs of the first-shell I–Ag distance (open squares) and second-shell I–I distance (open triangles) in β -AgI as a function of temperature (Daiba *et al.*, 1990). Absolute values have been obtained by fitting the slope of the experimental points to correlated Einstein models. Full squares and full triangles are the MSDs for the I–Ag and I–I pairs, respectively, calculated from the XRD data of Yoshiasa, Koto, Kanamaru, Emura & Horiuchi (1987).

Figure R1. Left: Debye-Waller factors obtained for the S-1_1(H₂) heating series. Right: data for AgI crystals

3(viii) There are procedures to determine the ss values and it is not just getting a good fit in R-space.

Reply to comment 3(viii)

We would be grateful to the reviewer if she/he can refer to any of those procedures which can be applied to materials in questions. Standard procedures are using the Einstein or Debye models (Scott Calvin, XAFS for everyone; <https://bruceravel.github.io/demeter/documents/Artemis/extended/ss.html>; <https://journals.aps.org/prb/pdf/10.1103/PhysRevB.20.4908>) but they apply only to well defined crystalline materials, often monoatomic (Citation from the Artemis/IFEFFIT user manual: “The caveat is that the correlated Debye model is only strictly valid for a monoatomic material. In practice, the Debye model works well for metals like Cu, Au, and Pt. It works poorly for any material that has two or more atomic species.”). Furthermore, these models do not predict in any way static disorder which would change in our sample upon temperature-induced desorption of water from the zeolite pores etc.

4. On page 6, it is suggested that the active site is binuclear. EXAFS is not a good method to determine if it is a monomolecular, bimolecular, trimolecular or other size. Single sites were previously proposed by poisoning stoichiometries for Zn+2 (ref 25) and EPR for Co+2 (ref 19). While it is possible that are exclusively binuclear sites as suggested, this conclusion is not supported by the data.

Reply to comment 4:

We agree with the reviewer that EXAFS is not a technique which can unambiguously and ultimately prove the nuclearity of clusters. It is nice that reference 25 (Ref. 26 in the revised manuscript) is mentioned as it is a good example how incomplete EXAFS evaluation can be used to mislead unexperienced referees and readers. One can see that in Ref. 25 (Ref. 26 in the revised manuscript) there is a small peak at uncorrected distance of 3 Å, the same as in our data (see Figure R2 below). Somehow the authors in Ref. 25 (Ref. 26 in the revised manuscript) do not mention this peak at all, do not even try to fit it, and arrive to a conclusion that there is no ordering beyond the first shell (or maybe no Zn, not discussed as said), i.e., single sites. Yet this peak at 3 Å is significant, it appears in spectra measured by different groups (thus not an artefact or noise) and more importantly and appears at a similar distance as in the ZnO spectrum. More than that, it fits perfectly to a Zn shell (and does not fit to a Si / Al shell), hence we believe that the conclusion about single sites by the authors of Ref. 25 (Ref. 26 in the revised manuscript) is because they overlooked

this contribution in EXAFS and none of the reviewers could spot the problem. For a contrast, one can also find in literature spectra of ZnOx species which do not show this peak at 3 Å (Figure R2, right, sample 4Zn/TiZrOx, copied from DOI 10.1021/acscatal.0c01580), those species are more likely to be attributed to single-site catalysts. Unfortunately, EXAFS fits or raw EXAFS data are not shown in Ref. 25 (Ref. 26 in the revised manuscript), hence it is difficult to discuss further.

Figure R2. Left – EXAFS data from Ref. 25 (Ref. 26 in the revised manuscript), middle – our work, right – exemplary EXAFS of ZnO_x single sites.

With respect to suitability of EXAFS to prove nuclearity, in spite of being not as specific as e.g., EPR (but Zn²⁺ is EPR silent) or Mössbauer spectroscopy, it is the only universal tool available to the community to study Zn species (Mössbauer and NMR are in principle possible but due to very low sensitivity are not used in practice). As said before, EXAFS strongly suggests a Zn-(O)-Zn coordination with a CN=1 giving a good fit. It is true that a CN=2 may still yield a reasonable fit with high but acceptable DW factor (SS) but that exhausts the possible options for the structural model. Hence, EXAFS is the only tool to obtain as good as reasonably achievable structural model and its use is widely accepted in the community exactly for this purpose. Most problems arise when it is mis-/abused as the example of the Ref. 25 (Ref. 26 in the revised manuscript) shows us. Thank you for the suggestion to probe the poisoning stoichiometry (titrate active sites), however it is also an indirect technique. It shows that each Zn atom is able to act independently, however it does not exclude the fact that another Zn atom is in the vicinity which EXAFS demonstrates. Thus, EXAFS and poisoning studies are providing different pieces of information as again Ref. 25 (Ref. 26 in the revised manuscript) exemplifies.

5. In ref 25, the Zn-O coordination number of the active catalyst decreased from 4 to 3 at high temperature in hydrogen as shown in this study. In 25, the true coordination of the latter thought to have 3 Zn-O and also contain a Zn-H, which would not be detected by EXAFS. In similar Ga PDH catalysts Ga-H has been detected by Raman and Infrared spectroscopy and the loss of Ga-O bonds correlates with Ga-H formation and not defect formation as suggested here.

Reply to comment 5:

First, we are very confused about how the referee knows that the single-sites Zn²⁺ in Ref.25 have 3 Zn-O and one Zn-H. We could not find this information in this reference (Ref. 26 in the revised manuscript). Those authors have written “For the microscopic reverse olefin hydrogenation pathway, H₂ heterolytically reacts at the Zn Lewis acid, producing a Zn–H and SiOH.” This statement does not mean that the active site must containing Zn-H bond. According to this previous paper, the structure of active site is (Si-O)₂-Zn-OH-Si.

Second, according to our in situ EXAFS spectra (Extended Data Figure 7 c), a gradual decrease in CN in the first Zn-O shell from 3 to 2 was established with an increase in reduction temperature (Extended Data Table 4). In other words, lattice oxygen was removed from binuclear Zn-oxo species. The removal should happen with participation of H₂. Consumption of oxygen in O₂-titration tests additionally proved the fact that lattice oxygen was removed upon reductive catalyst treatment.

Third, when CO was used as a reducing agent for catalyst preparation from ZnO and silicalite-1 according to Extended Data Figure 2a, the treated zeolite also shows high activity in PDH. Please see

Figure R3. Those catalysts show lower catalytic performance than their H₂-treated counterparts due to the low concentration of CO (1% CO vs. 50% H₂).

Figure R3 The rate of propene formation over S-1_1(CO) samples. Those samples (downstream located silicalite-1 (S-1_1(CO))) were prepared according to Extended Data Figure 2a at 550 °C, 600 °C and 650 °C. A mixture containing 1 vol% CO and 4 vol% Ne in Ar was used for reducing purposes for 2 h. Reaction conditions: 50 mg catalysts, 40 ml/min, C₃H₈:N₂=2:3, 550 °C.

Based on the abovementioned experimental observation, we can exclude the presence of Zn-H fragment in the active site.

We agree with the referee that Ga-H (*ACS Catal.* 2018, 8, 6106–6126; *J. Am. Chem. Soc.* 2018, 140, 4849–4859.) and In-H (*J. Am. Chem. Soc.* 2020, 142, 10, 4820–4832) could be formed during reduction.

6(i). The EPR in this study is suggested to results form a free electrons trapped at a reduced Zn vacancy site. The EPR signal is present in the as prepared. sample, which should have not vacancies.

Reply to comment 6(i)

The previous version contained an explanation for the F⁺ signal in the as prepared sample. “Anion vacancies were formed upon catalyst preparation under H₂ and only partially refilled after exposure to air at room temperature.” In order to avoid any misunderstanding, the present manuscript has been emended as follows. “In situ EPR (electron paramagnetic resonance) spectrum of fresh S-1-1_1(H₂) is characterized by a weak signal at g-factor of 2.005 characteristic for a trapped electron in an oxygen vacancy (F⁺ centre, Extended Data Figure 11). Anion vacancies were formed upon catalyst preparation under H₂ and only partially refilled after exposure to air at room temperature. The intensity of F⁺ signal increased after an additional exposure of S-1_1(H₂) to H₂/Ar=1 at 550 °C. The increase should be related to the formation of further F⁺ centres through removal of oxygen from binuclear Zn-oxo species. No F⁺ centres were found in the reduced bare S-1_1 support (Extended Data Figure 11). The presence of oxygen vacancies in reduced S-1_1(H₂) was also supported by O₂-titration tests at 550°C. This catalyst consumed O₂ until all vacancies were filled (Extended Data Figure 12).”

6(ii)The signal increase with the preparation treatment. However, the number of sites is not quantitatively compared to the intensity gain in the EPR. The EPR single has not been modeled to identify where the unpaired electron is located. Is it from the Zn ion, O ions, supports, adjacent to the two Zn if it is binuclear?
Reply to comment 6(ii)

Firstly, we should mention the details of EPR experiments. The catalyst prepared according to Extended Data Figure 2a was used for EPR experiment. This sample was initially measured at 20 °C. After reduction of this sample in the same set-up in H₂/N₂=1/1 at 550 °C for 1 h and cooling down to 20 °C, EPR spectrum was again recorded. Thus, we can safely conclude if the treatment caused changes in the

concentration of EPR active species. The experiment details have already been written in previous text (see line 524-529).

Secondly, we observed a signal with the g-factor of 2.005 in the S-1_1(H₂) sample containing binuclear Zn-oxo species. This signal is characteristic for an unpaired electron located in an oxygen vacancy (F⁺ centre). No other signals, which could be assigned to Zn⁺ or O⁻, were observed. As our *in situ* EXAFS results show that the CN of Zn-O decreases from 3 to 2 upon H₂ reduction from 100 °C to 500 °C, the F⁺ centre should be adjacent to the two Zn cations.

6(iii) It seems more likely that the signal is from the larger number of ZnO sites that are formed as the bulk oxide is dispersed on the support. It is not obvious that these are defect sites from the spectrum.

Reply to comment 6(iii)

Maybe we were not clear with the description of experimental details of EPR measurements. We did not use a physical mixture of ZnO and S-1 for the measurements. Otherwise, it would be impossible to explain the results in the presence of bulk ZnO and binuclear Zn-oxo species. The catalyst (S-1_1(H₂)) was prepared according to Extended data Figure 2a. ZnO and S-1_1 was separated by a quartz wool to avoid any physical contact. Thus, the S-1_1 sample exclusively possesses binuclear Zn-oxo species and O vacancies can only originate from binuclear Zn-oxo-species. (see our reply to comment 6(ii))

7. On page 8 it is suggested that OH group activate H₂. H-D exchange of support OH sites by D₂ on Pt catalysts have long been thought to be due to H atom spillover from the metal. However, careful studies by the Bourdard group have shown that if one traps the D₂O, which is present in the D₂, at liquid N₂ temperatures, there is no support OH-OD exchange, and support OH groups do NOT activate H₂. The IR in this study does not indicate that they removed all the D₂O present in the D₂. Thus, the OD peaks are likely due to exchange with D₂O, rather than D₂.

Reply to comment 7:

We did not suggest that OH groups activate H₂ or D₂. Contrarily, we wrote "In comparison with bare support, HD was observed upon pulsing of D₂/Ne=1 over S-1_1(H₂) (Extended Data Figure 14c)." Based on the comment of the reviewer, we assume that our description was not clear enough and provoked the criticism. We have modified the manuscript as follows. "Catalyst ability for hydrogen activation was elucidated by D₂-pulse experiments at 550°C. No isotopic exchange occurred over bare support. Contrarily, HD was observed upon pulsing of D₂/Ne=1 over S-1_1(H₂) (Extended Data Figure 14c). Thus, binuclear Zn-oxo species in this material must be involved in the exchange reaction. D atoms formed from D₂ react with OH groups of S-1_1 to yield HD. The latter also dissociates on binuclear Zn-oxo species to H and D. The H species react with another OH group to form gas-phase H₂. This explanation is supported by the fact that no H/D exchange was observed after all OH groups were converted into OD groups (Extended Data Figure 15). As the H/D exchange is a fast process and the amount of propene decreases when replacing C₃H₈ by C₃D₈, we put forward that the breaking of C–H bonds should be the rate-limiting step in propane dehydrogenation."

Additionally, please note that we did not use IR spectroscopy to analyse the products in the H/D exchange experiment. We used an online mass spectrometer (see "Method" in the main text).

8(i). Page 9 makes the case that these catalysts have potential for commercial applications. The Catofin process is a large capital-intensive unit, and the largest cost is separation of propylene from propane. The catalyst operates at the maximum possible temperature to obtain the highest conversion (above 50%) to minimize separation cost. This endothermic reaction leads to temperature drops and the conversion is quenched due to the equilibrium at the outlet temperature. Since the catalyst operates at equilibrium, higher rates do not lead to improved performance and higher yields. With the CrOx catalyst operation at higher T leads to lower selectivity. Process improvement to give higher space time yields and selectivity have been obtained by adding a heat generating material in the catalyst to increase the conversion while keeping the lower operating temperature. The CrOx catalysts has sufficient rate to give higher space time yields. Equilibrium limits the space time yields commercially. The limiting factors are selectivity and maximum operating temperature. The endothermic temperature drop controls the yields that any catalyst can get. In

this paper, at 23% conversion a selectivity of 90-92 is lower than the CrOx catalyst. This alone would not lead to commercial application.

Reply to comment 8(i)

We agree that higher reaction rates will not enable to overcome the thermodynamically limited degree of propane conversion. However, the higher the rate, the lower the catalyst amount is required to reach a certain conversion degree. Thus, having a highly active catalyst provides a possibility to reduce capital costs owing to reducing reactor volume.

To convince the reviewer about the application potential of our catalysts, we carried out a test for about 400 h on stream at 550°C using different feeds (40 or 70 vol% C₃H₈ and H₂/C₃H₈ of 0, 0.5 or 1) representative for the current large-scale PDH processes. An analogue of commercial K-CrO_x/Al₂O₃ and a catalyst consisting of S-1_3(H₂) with an upstream-located ZnO layer (ZnO//S-1_3(H₂)) were tested in parallel in the same set-up. To achieve comparable degree of propane conversion, 0.18 g ZnO//S-1_3(H₂) (0.09 g of ZnO and 0.09 g of S-1_3(H₂)) and 0.3018 g K-CrO_x/Al₂O₃ were used for the tests. The conversion of propane, and space time space of propene are shown in Figure R4 or Figure 3 in the main text in the emended manuscript. A decrease in the conversion of propane over ZnO//S-1_3(H₂) after about 250 h on propane stream is due to complete consumption of the ZnO layer. However, the initial conversion could be completely recovered after we added fresh ZnO on top of S-1_3(H₂).

Figure R4 Time on stream profiles of (a) propane conversion and (b) space time yield of propene (STY(C₃H₆)) obtained over ZnO-S-1_3 (blue) and an analogue of commercial K-CrO_x/Al₂O₃ (grey) under industrially relevant conditions. Reaction conditions: 550°C (0): C₃H₈:H₂:N₂=4:2:4, total flow was 10 ml/min or 6 ml/min, (I): C₃H₈:H₂:N₂=4:2:4, 6 ml/min, (II) C₃H₈:N₂=4:6, 6 ml/min, (III) C₃H₈:H₂:N₂=4:4:2, 6 ml/min, and (IV) C₃H₈:N₂=7:3, 6 ml/min. For the tests under (IV) conditions, weigh hourly space velocity (WHSV(C₃H₈)) for ZnO-S-1_3 and K-CrO_x/Al₂O₃ was 5.5 h⁻¹ and 1.64 h⁻¹, respectively. For all other conditions, the respective WHSV(C₃H₈) values were 3.1 h⁻¹ and 0.94 h⁻¹.

As seen in Figure R4, when we did not co-feed H₂, the propane conversion over ZnO-S-1_3 and K-CrO_x/Al₂O₃ was 38% and 39%, respectively. These values are close to the equilibrium conversion of 42% under the same conditions (*Chem. Soc. Rev.*, 2021, 50, 473). Under such conditions, our catalyst showed about 3 times higher space time yield of propene (Figure R4b or Figure 3b in the main text) with slightly lower selectivity to propene (on average 91.9% vs. 93.1%). However, our catalyst produces lower amounts of cracking products (Extended Data Figure 30) that might be advantageous for downstream distillation processes. Concerning undesired coke production, this undesired product in the CATOFIN process is combusted upon catalyst regeneration thus providing heat required for the endothermic PDH reaction.

8(ii). The reaction temperatures are also at 550C, which is about 75C too low for commercial operations. At higher T one gets more thermal cracking, so this Zn catalyst would be expected to have even lower selectivity. If the catalyst can operate at 650C, 50+% conversion with a selectivity of greater than 95% (and be regenerated 30,000 times), then this might replace the CrOx catalyst, however, the increase would be approximately what the CrOx catalyst is and not represent a step change in technology. For PDH, the rate of the catalyst is NOT the limiting factor of performance.

Reply to comment 8(ii)

We cannot agree with the referee's claim that "At higher T one gets more thermal cracking, so this Zn catalyst would be expected to have even lower selectivity". We tested ZnO-S-1_3 catalyst at 550 °C and 600 °C with different WHSV(C₃H₈). As seen in Figure R5, propane conversion was about 30% and 40% (see exact values in Extended Data Tables 6-7 in the revised Supplementary Information) at 550 °C and 600 °C, respectively. The selectivity to cracking products (methane and ethene) in these tests is similar (1.6% vs.1.9% for 550 °C vs. 600 °C). It also should be noticed that we compare them at different degree of propane conversion (~30% vs. ~40%). Additionally, some catalysts have already been reported to increase propene with an increase in the reaction temperature (*Applied Catalysis A, General*, 602(2020),117731). It depends on the activation energies of the dehydrogenation and cracking reaction.

Here, we would like to correct the referee, it is never a challenge to get high propene selectivity in PDH. The selectivity to propene and the conversion of propane highly depends on the feed composition at a certain reaction temperature. We could just use the same feed composition as shown in the Ref. 25 and Ref 19 (Ref. 26 and Ref. 20 in revised manuscript), which were mentioned many times by the referee in this comment, for example, 3 vol% propane in Ar. With such low amount of propane, they got above 95% selectivity to propene at 550 °C, 600 °C or 650 °C at the degree of propane conversion below 10% (we even could get 99% propene selectivity at such degree of propane conversion with 40% propane in N₂ at 550 °C). According to reviews (*Chem. Rev. 2014, 114, 10613–10653; Chem. Soc. Rev., 2021,50, 473–527*), such low concentration of propane in feed will result in high conversion and high selectivity to propene. Based on such knowledge, we have to remind the referee to notice that we used 40 vol% propane in all the tests! So, the real challenge for PDH is how to get high selectivity at high propane conversion with high concentration of propane (at least>30vol%).

Finally, we should highlight again that the advantages of our work are (i) the preparation method (to show how to prepare catalysts by a simple method which could be used for large-scale production) and (ii) the different function of different OH groups on the SiO₂-based supports (to show how and why the different OH groups could form different Zn-oxo species). Such knowledge could guide researchers to develop more active species and catalysts.

Figure R5 The conversion of propane and selectivity to propene over ZnO-S-1_3 mixture without H₂ cofeed at different temperatures. a 550 °C, C₃H₈:N₂=2:3, the WHSV(C₃H₈) is 7.9 h⁻¹, b 600 °C, C₃H₈:N₂=2:3, the WHSV(propene) is 15.7 h⁻¹. It had already been shown in the previous Supplementary Information. Please see exact values in Extended Data Tables 6-7 in the revised Supplementary Information.

(9).The materials reported in this manuscript don't appear to be much different from those previously reported for Zn PDH catalysts. Many of the conclusions are known or not supported by the results, thus this is paper is not recommended.

Reply to comment 9.

From a chemical viewpoint, our Zn-containing catalysts do not differ from those reported in previous studies but show significantly higher activity in comparison with previously tested Zn-containing catalysts and other state-of-the art materials even containing Pt. Nevertheless, we do not highlight the catalyst. The primary advantages of our study are (i) a simple method for preparation of highly active ZnO-containing catalyst based on ZnO and commercially available oxidic supports and (ii) the fundamentals enabling preparation of catalysts in a purposeful manner (for details see our reply to your general statement).

We hope that our thorough reply to the specific comments will change the opinion of the reviewer about the novelty of our study and the conclusions made.

Referee #2 (Remarks to the Author):

A. In my opinion, the authors have significantly improved the manuscript since its first iteration. The manuscript is now cohesive to read between different sections, some additional data was added and critical issues have been addressed.

Reply to comment A:

Thank the referee for her/his high evaluation of our study.

B. The work is of high significance as it pertains to the highest volume chemicals manufactured worldwide. The work is reasonably original and the findings would be of interest to the scientific community in industrial chemistry, chemical engineering and materials characterization.

Reply to comment B:

Thank the referee for her/his high evaluation.

C. In this reviewer's opinion, the standout feature of the article is the thorough characterization methods for the catalyst and surface intermediates using excellent characterization instrumental and experimental tools. The validity of the approach is excellent, the data quality is sufficient and the presentation is easier to follow. The integrated DFT computational mechanics models and discussion contribute to the value of the manuscript.

Reply to comment C:

Thank the referee for her/his high evaluation.

D. The use of statistics and treatment of uncertainties appears acceptable.

Reply to comment D:

Thank the referee for her/his high evaluation.

E. It appears that the authors have put together an impressive multi-institutional effort to resolve a complicated scientific problem. In my opinion, the wealth of data was put to good use and the conclusions are well supported. This reviewer did not notice issues that would cause validity or reliability concerns.

Reply to comment E:

Thank the referee for her/his high evaluation.

F. No. The original manuscript had some serious issues that have been well addressed in this revision.

Reply to comment F:

Thank the referee for her/his high evaluation.

G. References on previous work on the subject are adequate.

Reply to comment G:

Thank the referee for her/his high evaluation.

H. The clarity and context are fine after the revision. This reviewer would prefer for the authors to improve upon the abstract a bit. For example, comments about the superiority of the studied catalyst do not really belong there, as this is still a laboratory, experimental system.

Note: I believe on line 36 the authors likely meant Cr(VI) instead of Cr(IV).

Reply to comment H:

Thank the referee for her/his suggestions. We have modified the abstract and carefully read the manuscript.

Reviewer Reports on the Second Revision:

===== Referees' comments =====

Referee #1 (Remarks to the Author):

While it is clear that this synthesis of Zn in silicalite S-1 is a good catalyst, still am not convinced of the structural characterization. The EXAFS data is high quality, but the results are not presented such that the fit results and conclusions that these lead to are correct.

1. It is suggested on page 6 that the Zn-O CN of ZnO is 2.5 (line 146). As giving in the SI text under SI T4, the Zn-O CN of ZnO is 4. The authors should show the fits in table for the reference ZnO. Is the CN 2.5 or 4? If the former, then the fits in Table S4 need corrections and will be higher than reported. The lower CN in some samples is suggested to be a mixture of ZnO and Zn; however, the higher shell distances are 3.25 Å, rather than 2.67 Å for that of Zn metal.

2. The Zn-O fit of Zn-S_1 is 3.4 and there is no fit of the higher shell peak. For the reduced Zn-S_1, there is a loss of Zn-O CN and shall higher shell peak, Fig 1e and fits in SI Table 4. To be convincing that this is a dimeric Zn-O-Zn, several things need to be shown.

a. The EXAF of the Zn-S_1 in air along with sample reduced in H₂ (100C, for example). Are these different? To confirm that these are different one could show subtract the two chi files and show the FT of the difference spectrum. The imaginary part of the FT should be compared to that of ZnO scaled to overlap with the small peak of the catalyst sample. Are these identical or different?

b. The higher shell peaks of the reduced are small and difficult to fit. The quality of the fits needs to be shown. The higher shell peak should also be modeled with a Zn-O-Si scatter. The bond distance can be taken from studies of Zn in ZSM-5, which have been reported. If the fit quality is not good, or can be modeled with Si, then the binuclear active site is not correct.

c. If the higher shell is consistent with ZnO, can the authors eliminate the possibility that the sample contains a small amount of ZnO and isolated Zn-S_1 sites? The other samples with catalytic activity presumably have some active sites and ZnO.

3. The Experimental has catalyst preparations of Ga₂O₃ and supports of Al₂O₃, Al-SiO, ZrO₂, etc. These are not reported. Ga would not be expected to be reduced to metallic Ga under these conditions to prepared equivalent Ga catalysts. The experimental should only include those catalysts, methods, etc., that are reported in the manuscript.

The method clearly leads to an active catalyst in silicalite_1. The other catalysts are not particularly active. The nature of the active site is difficult to determine exactly, but if the bimolecular active site is much more active than an isolated site, this is important. Thus, critical analysis to confirm this conclusion is required. I'm not convinced that this has be done yet.

Referee #2 (Remarks to the Author):

In a previous review of the work, I recommended this work for publication after a number of issues was addressed thoroughly by the authors.

A reread confirms that this work is now well assembled with a good justification of the conclusions from the data.

Overall, I recommend this manuscript for publication.

===== End of comments =====

Author Rebuttals to Second Revision:

Reply to the comments:

The scientific comments from the referees are highly appreciated by the authors. The manuscript has been revised according to the style and format requirements of *Nature*. Our detailed replies to the specific comments are given below.

Referee #1 (Remarks to the Author):

While it is clear that this synthesis of Zn in silicalite S-1 is a good catalyst, still am not convinced

of the structural characterization. The EXAFS data is high quality, but the results are not presented such that the fit results and conclusions that these lead to are correct.

Reply to the general statement:

We thank the referee for her/his high evaluation of the potential of our ZnO-silicalite-1 catalyst and the quality of EXAFS data. We were probably not enough precise upon presentation and discussion of the characterization results and provoked the criticism of the referee. We have applied the state-of-the-art approaches upon evaluating the EXAFS data. We hope that our below point-by-point reply to the specific comments of the referee will change her/his opinion about the correctness of the fit results and the conclusions drawn on this basis.

Comment 1(a)

It is suggested on page 6 that the Zn-O CN of ZnO is 2.5 (line 146). As giving in the SI text under SI T4, the Zn-O CN of ZnO is 4. The authors should show the fits in table for the reference ZnO. Is the CN 2.5 or 4? If the former, then the fits in Table S4 need corrections and will be higher than reported.

Reply to comment 1(a)

We thank the referee for this comment. As described in the Methods section, “the amplitude reduction factor $S_0^2=1.08$ was obtained by fitting the ZnO reference spectrum to a wurtzite structural model”. This means that the fitting was done upon fixing the coordination numbers to the tabulated values, i.e., the Zn-O CN in the first shell was set to 4. Thus, S_0^2 determined from this fit was used for analyzing the spectra of other catalysts. We have provided the ZnO fit in the Figure R1 or Extended Data Figure 5c-d.

Figure R1 EXAFS fits of the ZnO reference (used to estimate amplitude reduction factor) in (a) r- and (b) q-spaces. ZnO fit summary (based on wurtzite structural model): $\delta E_0 = 1.0 \pm 2.6$ eV; $\rho = 2.0\%$; amplitude reduction factor (S_0^2) = 1.08 ± 0.2 ; $\sigma^2 = 10.1 \pm 1.8 \times 10^{-3} \text{ \AA}^2$ (the same for all fitted shells). First shell comprises 4 O atoms at $1.96 \pm 0.03 \text{ \AA}$; second shell: 1 O atom at $3.11 \pm 0.34 \text{ \AA}$, 6 Zn atoms at $3.21 \pm 0.02 \text{ \AA}$ and 6 Zn atoms at $3.26 \pm 0.02 \text{ \AA}$.

Comment 1(ii)

The lower CN in some samples is suggested to be a mixture of ZnO and Zn; however, the higher shell distances are 3.25 Å, rather than 2.67 Å; for that of Zn metal.

Reply to comment 1(ii)

We apologize for a possible misunderstanding. We have originally written in the manuscript "... MCM-41(H₂) should contain partially reduced ZnO_x nanoparticles". This sentence does not necessarily mean that the catalyst contains a mixture of Zn and ZnO. To avoid any misunderstanding, we rewrote this part as follows "Partially reduced ZnO_x (defect structure with oxygen vacancies) should be present in MCM-41(H₂) (Extended Data Figure 4a in the revised manuscript)".

Comment 2a.

The Zn-O fit of Zn-S_1 is 3.4 and there is no fit of the higher shell peak. For the reduced Zn-S_1, there is a loss of Zn-O CN and shall higher shell peak, Fig 1e and fits in SI Table 4. To be convincing that this is a dimeric Zn-O-Zn, several things need to be shown.

a. The EXAF of the Zn-S_1 in air along with sample reduced in H₂ (100C, for example). Are these different? To confirm that these are different one could show subtract the two chi files and show the FT of t^oChe difference spectrum. The imaginary part of the FT should be compared to that of ZnO scaled to overlap with the small peak of the catalyst sample. Are these identical or different?

Reply to comment 2a

The spectra of S-1_1(H₂) in air and in H₂ at 100 °C (at higher temperatures comparison is more difficult because of the changed Debye-Waller factor) are identical (except for the noise of course) and there is fit of the higher shell peak. They can be seen in the revised Extended Data Figure 5a and b. The EXAFS fit parameters are given in Extended Data Table 4.

Comment 2b (i)

The higher shell peaks of the reduced are small and difficult to fit. The quality of the fits needs to be shown.

Reply to comment 2b (i):

Comparison of the fits and the actual data in the R-space (magnitude and real part, imaginary not shown as it is complementary to the real and just overloads the figures without additional value) and q-space (fits+Fourier-filtered data) as well as all fit details (including the lack of fit ρ) are given in Extended Data Figures 6-7 and Extended Data Table 4. As this information had already been provided in the first-revision round (e.g. **Specific comment 1 and Specific comment 14**), we are quite unsure either we correctly understand the concern of the referee or our reply is not sufficiently convincing. If the latter, please, explain why.

Comment 2b (ii)

The higher shell peak should also be modeled with a Zn-O-Si scatter. The bond distance can be taken from studies of Zn in ZSM-5, which have been reported. If the fit quality is not good, or can be modeled with Si, then the binuclear active site is not correct.

Reply to comment 2b (ii):

In comparison with the binuclear ZnO_x model, using Si instead of Zn is not able to satisfactorily model the second shell as seen in Figure R2. The corresponding fit parameters are given in Table R1. This table and Figure R2 are now available as Supplementary Information Table 1 and Extended Data Figure 8, respectively.

Table R1 *Ex situ* EXAFS fit parameters of S-1_1(H₂) with either Zn or Si in the second shell

Element in the 2 nd shell	Zn-O distance (Å)	CN (Zn-O)	σ^2 for O (10 ⁻³ Å ²)	Zn-Zn/Si distance (Å)	CN(Zn-Zn/Si)	δE_0 (eV)	ρ (%)
Fit with Zn	1.97±0.01	2.9±0.2	7.3±1.6	3.32±0.07	1.3±0.5	4.7±0.6	0.2
Fit with Si	1.95±0.01	3.2±0.2	7.5±1.2	3.12±0.07	0.4±0.3	2.1±0.9	0.2

Figure R2. Fitting of the *ex-situ* spectrum of the S-1_1(H₂) sample using Si or Zn in the second shell presented in (a), (c) and (b), (d) for r- and q-spaces. A significant misfit can be seen for

the second shell peak at approx. 3.0 Å.

Comment 2c

If the higher shell is consistent with ZnO, can the authors eliminate the possibility that the sample contains a small amount of ZnO and isolated Zn-S₁ sites? The other samples with catalytic activity presumably have some active sites and ZnO.

Reply to 2(c)

We hope that the below arguments convince the referee that the coexistence of isolated ZnO_x sites and a small amount of ZnO nanoparticles in S-1_1(H₂) can be eliminated. The S-1_1(H₂) should possess unique ZnO_x sites differing in their structural characteristics and intrinsic catalytic activity from isolated ZnO_x species and ZnO nanoparticles.

i) Based on the fit results, the Zn-Zn distance in ZnO_x species in S-1_1(H₂) is larger than in ZnO, i.e. 3.32 Å vs 3.213 or 3.25 Å.

ii) We prepared Figure R3 directly comparing the S-1_1(H₂) XANES spectrum with those of ZnO and a previously reported spectrum of single Zn O_x species on a different support [ACS Catal. 2020, 10, 8933–8949]. The latter completely lacks the backscattering peak at ca. 3.0 Å (uncorrected distance). While the EXAFS region (high energy oscillation) is very similar for single and binuclear sites (dominated by the first shell, which has 3 O atoms in both cases), the edge regions of the spectra are markedly different. More importantly, the absence of isosbestic points in the given dataset shows that S-1_1(H₂) spectrum cannot be thought as a mixture of ZnO and single ZnO_x sites. In other words, there is no possible way to combine the blue (isolated ZnO_x) and the red (ZnO) spectra to obtain the black spectrum (S-1_1(H₂)).

Figure R3. *Ex-situ* XANES (including the first EXAFS oscillation) spectra of the S-1_1(H₂) sample together with reference data for ZnO and supported single Zn sites [ACS Catal. 2020, 10, 8933–8949].

iii) To demonstrate that the S-1_1(H₂) sample possesses peculiar active sites, we compare the TOF values obtained at 550 °C over this catalyst (present study), single-site Zn²⁺/SiO₂ (ACS Catal. 4, 1091-1098 (2014)), ZnO NPs/MCM-41 (present study) and single site Zn²⁺/TiZrO_x (ACS Catal. 10, 8933-8949 (2020)). The TOF value of S-1_1(H₂) is about 300, 100 or 2 times higher than that of Zn²⁺/SiO₂, ZnO NPs/MCM-41 or single site Zn²⁺/TiZrO_x (Figure R4 or Figure 1h in the emended manuscript). If our catalyst had a mixture of ZnO and isolated Zn²⁺,

its activity would be between that of ZnO NPs/MCM-41 and single-site Zn²⁺/SiO₂ or Zn²⁺/TiZrO_x, i.e., between 2.52 h⁻¹ and 0.77 h⁻¹ or between 2.52 h⁻¹ and 105 h⁻¹.

Figure R4 Turnover frequency (TOF) of propene formation over different Zn-based catalysts. The TOF values of single Zn²⁺/SiO₂ and single Zn²⁺/TiZrO_x are reported in Ref. 19 and Ref. 24 in the revised manuscript, respectively.

[Redacted]

iv) We have also performed additional DFT calculations of the PDH reaction on single ZnO^{2+} supported on S-1. In comparison with the binuclear ZnO_x species, the whole energy span and the apparent activation barrier are higher indicating single $\text{ZnO}_x^{2+}/\text{S-1}$ should be much less active. (Figure R6, Figure 2, Extended Data Figures 12-13 and Supplementary Information Figures 2-4).

Figure R6 Molecular details of propane dehydrogenation. The calculated energy profiles along the minimum energy pathways of propane dehydrogenation to propene on reduced binuclear Zn-oxo species and single Zn²⁺/S-1 on the surface of S-1 and the optimized structures(A-C) of intermediates and transition states (TS) at the PBE+D3+ZPE level. Cyan, red, grey, yellow and white stand for Zn, O, C, Si and H, respectively.

In summary, based on EXAFS results and their evaluation as suggested by the referee, additional catalytic tests and DFT calculations, we can conclude that ZnO_x/S-1 should possess binuclear Zn-oxo species rather than a mixture of isolated ZnO_x and ZnO_x nanoparticles.

Comment 3

The Experimental has catalyst preparations of Ga₂O₃ and supports of Al₂O₃, Al-SiO, ZrO₂, etc. These are not reported. Ga would not be expected to be reduced to metallic Ga under these conditions to prepared equivalent Ga catalysts. The experimental should only include those catalysts, methods, etc., that are reported in the manuscript.

Reply to comment 3:

We agree with the referee that Ga₂O₃ is not expected to be reduced under H₂ conditions. A physical mixture of Ga₂O₃ and silicalite-1, however, shows much high propene formation rate (0.645 mmol g⁻¹ min⁻¹) than the bare support (0.008 mmol g⁻¹ min⁻¹) and Ga₂O₃ (0.097 mmol g⁻¹ min⁻¹). Based on a recent article published in *Nature* (2020, volume 585, pages 221–224 (2020)) dealing with Pt and rare-earth or transition metal oxides, we cannot exclude that OH

necks can participate in the reduction of Ga₂O₃. To avoid any not 100% proven conclusions, we have decided to omit discussion of Ga-containing catalysts from the emended manuscript. We continue to work on this topic.

General statement

The method clearly leads to an active catalyst in silicalite_1. The other catalysts are not particularly active. The nature of the active site is difficult to determine exactly, but if the bimolecular active site is much more active than an isolated site, this is important. Thus, critical analysis to confirm this conclusion is required. I am not convinced that this has been done yet.

Reply to the general statement:

We are quite unsure why the referee has concluded “The other catalysts are not particularly active”. Even the original manuscript (first submission) contains other than silicalite-1-based catalysts showing similar or even superior activity to the catalyst mentioned by the referee. Figure 3c, d in emended manuscript also shows several catalysts with higher activity in comparison with an analogue of commercial K-CrO_x/Al₂O₃ catalyst (see grey line in Figure 3c and d in the emended manuscript). Importantly, our preparation method works with any zeolite or metal oxide if they possess defective OH groups.

Concerning the nature of the active sites, please, see our point-by-point reply to your specific comments 1-2.

Referee #2 (Remarks to the Author):

In a previous review of the work, I recommended this work for publication after a number of issues was addressed thoroughly by the authors. A reread confirms that this work is now well assembled with a good justification of the conclusions from the data.

Overall, I recommend this manuscript for publication.

Reply to Referee 2#:

We sincerely appreciate Referee 2 for her/his positive evaluation of our work.

Reviewer Reports on the Third Revision:

===== Referees' comments =====

Referee #1 (Remarks to the Author):

The EXAFS analysis is more complete and I'm happy with the authors response and changes in the paper.

I recommend publication.

===== End of comments =====